# Efficient Prior Selection in Gaussian Process Bandits with Thompson Sampling

## Abstract

Gaussian process (GP) bandits provide a powerful framework for performing blackbox optimization of unknown functions. The characteristics of the unknown function depend heavily on the assumed GP prior. Most work in the literature assume that this prior is known but in practice this seldom holds. Instead, practitioners often rely on maximum likelihood estimation to select the hyperparameters of the prior - which lacks theoretical guarantees. In this work, we propose two algorithms for joint prior selection and regret minimization in GP bandits based on GP Thompson sampling (GP-TS): Prior-Elimination GP-TS (PE-GP-TS) and HyperPrior GP-TS (HP-GP-TS). We theoretically analyze the algorithms and establish upper bounds for the regret of HP-GP-TS. In addition, we demonstrate the effectiveness of our algorithms compared to the alternatives through experiments with synthetic and real-world data.

## 1 Introduction

In Gaussian process bandits, we consider a variant of the multi-armed bandit problem where the arms are correlated and their expected reward is sampled from a Gaussian process (GP). The flexibility of GPs have made GP bandits applicable in a wide range of areas that need to optimize blackbox functions with noisy estimates, including machine learning hyperparameter tuning (Turner et al., 2021), drug discovery (Hernández-Lobato et al., 2017; Pyzer-Knapp, 2018), online advertising (Nuara et al., 2018), portfolio optimization (Gonzalvez et al., 2019) and energy-efficient navigation (Sandberg et al., 2025). Most of the theoretical results in the literature assume that the GP prior is known but this is seldom the case in practical applications. Even with expert domain knowledge, selecting the exact prior to use can be a difficult task. Most practitioners tend to utilize maximum likelihood estimation (MLE) to identify suitable prior parameters. However, in a sequential decision making problem MLE is not guaranteed to recover the correct parameters.

In the literature, Wang & de Freitas (2014); Berkenkamp et al. (2019); Ziomek et al. (2024) provided algorithms with theoretical guarantees when the kernel lengthscale is unknown. More recently, Ziomek et al. (2025) introduced an elimination-based algorithm with theoretical guarantees for an arbitrary set of discrete priors. Their algorithm, Prior-Elimination GP-UCB (PE-GP-UCB), selects the arm and prior which provide the most optimistic upper confidence bound (UCB). If a prior generates too many incorrect predictions, then it may be eliminated. The previous work has focused on optimistic UCB methods which are known to over-explore.

In this work, we investigate the use of Thompson sampling for solving GP-bandit problems with unknown priors and we propose two algorithms. The first algorithm, Prior-Elimination GP-TS (PE-GP-TS), is an extension of PE-GP-UCB that replaces the doubly optimistic selection rule with posterior sampling and one less layer of optimism. We analyze the regret of PE-GP-TS. For the terms we can bound, we obtain a regret bound for PE-GP-TS of order $\mathcal{O}(\sqrt{T \log T |P| \hat{\gamma}_T})$ where $T$ is the horizon, $|P|$ is the number of priors and $\hat{\gamma}_T$ is the worst-case maximum information gain, which matches that of PE-GP-UCB. The second algorithm, HyperPrior GP-TS (HP-GP-TS), uses bi-level posterior sampling to efficiently explore the priors and arms. An UCB-based analysis of HP-GP-TS yields a regret bound of order $\mathcal{O}(\sqrt{T \log T \bar{\gamma}_T})$ (where $\bar{\gamma}_T$ is the average maximum information gain) plus a term that corresponds to the cost of learning the optimal prior. An information-theoretic analysis yields a regret bound of order $\mathcal{O}(\sqrt{T |\mathcal{X}| \log |\mathcal{X}|})$ where $|\mathcal{X}|$ is the number of arms.

We evaluate our methods on three sets of synthetic experiments and three experiments with real-world data. Across the experiments, our Thompson sampling based methods outperform PE-GP-UCB. Additionally, we find that the regret of HP-GP-TS does not increase with $|P|$ in our experiments. Finally, we analyze the priors selected by the algorithms and observe that HP-GP-TS selects the correct prior more often than the other algorithms.

The contributions of this work can be summarized as:

- We propose two novel algorithms for GP-bandits with unknown prior: PE- and HP-GP-TS.
- We theoretically analyze the regret of HP-GP-TS using a UCB framework and an information-theoretic framework which provides a regret bound of order $\mathcal{O}(\sqrt{T|\mathcal{X}|\log|\mathcal{X}|})$. Additionally, we analyze the regret of PE-GP-TS.
- We experimentally evaluate our algorithms on both synthetic and real-world data, demonstrating that they achieve superior performance and that the regret of HP-GP-TS does not increase with $|P|$.

## 2 BACKGROUND AND PROBLEM STATEMENT

**Problem statement**   We consider a sequential decision making problem where an agent repeatedly selects among a set of arms and receives a random reward whose mean depends on the selected arm and is unknown to the agent. The goal of the agent is to maximize the cumulative sum of rewards over a finite time horizon. We assume that the distribution of the means, the *prior*, is sampled from a set of priors, the *hyperprior*. An effective agent must distinguish which prior the means are sampled from to ensure it explores efficiently.

Now, let us formally state the problem. Let $\mathcal{X} \subseteq [0, r]^d \subset \mathbb{R}^d$ denote the finite set of arms and $P$ a finite set of priors with associated prior mean and kernel functions $\mu_{1,p} : \mathcal{X} \mapsto \mathbb{R}$ and $k_{1,p} : \mathcal{X} \times \mathcal{X} \mapsto \mathbb{R}, \forall p \in P$. Let $p^* \in P$ denote the true prior and assume the expected reward function $f : \mathcal{X} \mapsto \mathbb{R} \sim \mathcal{GP}(\mu_{1,p^*}, k_{1,p^*})$ is a sample from a Gaussian process with prior $p^*$. Both the function $f$ and the true prior $p^*$ are considered unknown. We will consider two settings: In the frequentist selection setting, the prior $p^* \in P$ is picked arbitrarily. In the Bayesian selection setting, the prior is sampled from a hyperprior $p^* \sim \mathcal{P}(P)$. To simplify notation, let $P_1$ denote the hyperprior.

Let $T$ denote the horizon. For time step $t = 1, 2, \ldots, T$, the agent selects an arm $x_t \in \mathcal{X}$ and observes the reward $y_t = f(x_t) + \epsilon_t$ where $\{\epsilon_t\}_{t=1}^T$ are i.i.d. zero-mean Gaussian noise with variance $\sigma^2$. The goal of the agent is to select a sequence of arms $\{x_t\}_{t=1}^T$ that minimizes the regret $R(T) = \sum_{t \in [T]} f(x^*) - f(x_t)$ where $[T] = \{1, \ldots, T\}$ and $x^* = \arg\max_{x \in \mathcal{X}} f(x)$. In the Bayesian selection setting, we evaluate the agent based on the Bayesian regret $\mathrm{BR}(T) = \mathbb{E}[R(T)]$ where the expectation is taken over the prior $p^*$, the expected reward function $f$, the noise $\{\epsilon_t\}_{t=1}^T$ and the (potentially) stochastic selection of arms.

**Gaussian processes**   A Gaussian process $f(x) \sim \mathcal{GP}(\mu, k)$ is a collection of random variables such that for any subset $\{x_1, \ldots, x_n\} \subset \mathcal{X}$, the vector $[f(x_1), \ldots, f(x_n)] \in \mathbb{R}^n$ has a multivariate Gaussian distribution. The probabilistic nature of GPs make them very useful for defining and solving bandit problems where the arms are correlated. Given the history $H_t = \{(x_i, y_i)\}_{i=1}^{t-1}$, the posterior mean and kernel functions of a Gaussian process $\mathcal{GP}(\mu, k)$ are given by

$$\mu_t(x) = \mu(x) + \mathbf{k}^\top \left(\mathbf{K} + \sigma^2 I\right)^{-1} (\mathbf{y} - \boldsymbol{\mu}), \tag{1}$$

$$k_t(x, \tilde{x}) = k(x, \tilde{x}) - \mathbf{k}^\top \left(\mathbf{K} + \sigma^2 I\right)^{-1} \tilde{\mathbf{k}}. \tag{2}$$

Above, $\mathbf{k}, \tilde{\mathbf{k}} \in \mathbb{R}^{t-1}$ are vectors such that $(\mathbf{k})_i = k(x_i, x)$ and $(\tilde{\mathbf{k}})_i = k(x_i, \tilde{x})$. Additionally, $\mathbf{y}, \boldsymbol{\mu} \in \mathbb{R}^{t-1}$ are also vectors such that $(\mathbf{y})_i = y_i$ and $(\boldsymbol{\mu})_i = \mu(x_i)$. The gram matrix is denoted by $\mathbf{K} \in \mathbb{R}^{t-1 \times t-1}$ where $(\mathbf{K})_{i,j} = k(x_i, x_j)$. Let $\mu_{t,p}$ and $k_{t,p}$ denote the posterior mean and kernel for a Gaussian process with prior $p \in P$ at time $t$ and let $\sigma_{t,p}^2(x) = k_{t,p}(x, x)$ denote the posterior variance at time $t$. The kernel $k$ determines important characteristics of the functions $f$, see Section B for more details and examples.

**Information gain**   The maximal information gain (MIG) is a measure of reduction in uncertainty of $f$ after observing the most informative data points up to a specified size. The MIG commonly occurs

in regret bounds for GP bandit algorithms (Srinivas et al., 2012; Vakili et al., 2021) and its growth rate is strongly determined by the prior kernel of the GP. Hence, we will define the MIG for any fixed GP prior $p \in P$. Let $\mathbf{y}_A$ denote noisy observations of $f$ at the locations $A \subset \mathcal{X}$. Then, the MIG given prior $p \in P$, $\gamma_{T,p}$, is defined as

$$\gamma_{T,p} := \sup_{A \subset \mathcal{X}, |A| \leq T} I_p(\mathbf{y}_A; f), \qquad (3)$$

where $I_p(\mathbf{y}_A; f) = H(\mathbf{y}_A|p) - H(\mathbf{y}_A|f, p)$ is the mutual information between $\mathbf{y}_A$ and $f$ given $p$, and $H(\cdot)$ denotes the entropy. To aid our analysis later, we also define the worst-case MIG as $\hat{\gamma}_T := \max_{p \in P} \gamma_{T,p}$ and the average MIG as $\bar{\gamma}_T := \mathbb{E}_{p \sim P_1}[\gamma_{T,p}]$. For the RBF and Matérn kernels, $\gamma_{T,p} = \mathcal{O}(\log^{d+1}(T))$ and $\gamma_{T,p} = \mathcal{O}(T^{\frac{d}{2\nu+d}} \log^{\frac{2\nu}{2\nu+d}}(T))$ (Srinivas et al., 2012; Vakili et al., 2021).

**Previous work** Plenty of previous work have proposed fully Bayesian approaches that integrate the acquisition function over the hyperposterior (Osborne et al., 2009; Benassi et al., 2011; Snoek et al., 2012; Hernández-Lobato et al., 2014; Wang & Jegelka, 2017; De Ath et al., 2021; Hvarfner et al., 2023). In contrast, HP-GP-TS optimizes a single hyperposterior sample instead of expected values over the hyperposterior.

Wang & de Freitas (2014) first derived regret bounds for GP bandits with unknown lengthscale for the Expected Improvement algorithm (Močkus, 1975). However, the proposed algorithm requires a lower bound on the lengthscale and the regret bound depends on the worst-case MIG. Later work by Berkenkamp et al. (2019) introduced Adaptive GP-UCB (A-GP-UCB) that continually lowers the lengthscale parameter. Given a sufficiently small lengthscale, the function $f$ lies within the reproducing kernel Hilbert space (RKHS) and the regular GP-UCB theory can be applied. However, A-GP-UCB lacks a stopping mechanism and will overexplore as the lengthscale continues to shrink. Recent work by Ziomek et al. (2025) introduced Prior-Elimination GP-UCB (PE-GP-UCB) for time-varying GP-bandits with unknown prior. Unlike the work before, the regret bound of PE-GP-UCB holds for arbitrary types of hyperparameters in the GP prior. PE-GP-UCB is doubly optimistic and selects the prior *and* arm with the highest upper confidence bound. PE-GP-UCB tracks the cumulative prediction error made by the selected priors and eliminates priors that exceed a threshold level.

Other works have introduced regret balancing algorithms that maintain a set of base learning algorithms and balance their selection frequency to achieve close to optimal regret (Abbasi-Yadkori et al., 2020; Pacchiano et al., 2020). Ziomek et al. (2024) built on this idea and introduced length-scale balancing GP-UCB which can adaptively explore smaller lengthscales but can return to longer ones, unlike A-GP-UCB.

The aforementioned works are based on UCB, EI, PI or regret balancing. However, another line of work has studied Thompson sampling in standard and linear bandits with unknown prior distribution (Kveton et al., 2021; Basu et al., 2021; Hong et al., 2022a; Li et al., 2024). In their setting (meta or hierarchical bandits), the agent plays multiple bandit instances, either simultaneously or sequentially. The unknown means are sampled from the same (unknown) prior and by gathering knowledge across instances, the agent can solve later instances more efficiently once it has identified the prior. Hong et al. (2022b) studied Thompson sampling with mixture priors (MixTS) where the agent only interacts with one instance. HP-GP-TS can be seen as an instantiation of MixTS for Gaussian process bandits. Hong et al. (2022b) provide the outline of a regret bound for a standard bandit setting and a detailed proof for the linear setting. However, we highlight some issues that we believe invalidate their analysis for the linear case in Section E. We emphasize that these methods have ~~been studied only for standard stochastic and linear bandits,~~ not been studied for GP bandits.

## 3 ALGORITHMS

As discussed by Russo & Van Roy (2014), TS can offer advantages over UCB algorithms for problems where constructing tight confidence bounds is difficult. In addition, Thompson sampling is often observed to perform better than UCB in practice (Chapelle & Li, 2011; Wen et al., 2015; Kandasamy et al., 2018; Åkerblom et al., 2023b;a). Motivated by this, we present two algorithms for efficient prior selection based on TS.

### 3.1 PRIOR-ELIMINATION WITH THOMPSON SAMPLING

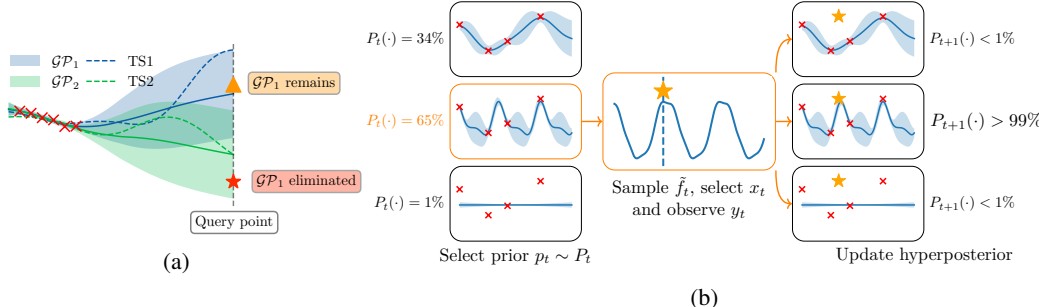

(a)

(b)

Figure 1: a) Elimination procedure of PE-GP-TS. The solid lines correspond to posterior means and the shaded regions are confidence intervals. The figure has been adapted from Ziomek et al. (2025). The dashed lines are samples from the posteriors. b) Overview of HP-GP-TS. The orange star corresponds to $y_t$.

Our first algorithm is an extension of PE-GP-UCB (Ziomek et al., 2025) to be employed with Thompson sampling - instead of UCB. The key difference is that instead of maximizing the upper confidence bound $U_t(x, p) = \mu_{t,p}(x) + \sqrt{\beta_t}\sigma_{t,p}(x)$ over $\mathcal{X} \times P_{t-1}$, we instead sample $\tilde{f}_{t,p}$ from the posterior $\mathcal{GP}(\mu_{t,p}, k_{t,p})$ for all priors $p \in P_{t-1}$ where $P_{t-1}$ is the set of active priors. Then, we select the arm and prior $x_t, p_t$ such that $x_t, p_t = \arg\max_{x,p \in \mathcal{X} \times P_{t-1}} \tilde{f}_{t,p}(x)$. Whilst PE-GP-UCB has two layers of optimism, the upper confidence bound and joint maximization of $x$ and $p$, PE-GP-TS has only a single layer of optimism - which should alleviate potential overexploration issues.

The elimination procedure of PE-GP-TS is illustrated in Fig. 1. Samples $\tilde{f}_{t,p}$ are drawn from the active prior $p \in P_{t-1}$. Then, the unknown function $f$ is queried at the selected arm $x_t$. If the observed value differs too much from the prediction made by the selected prior, then the selected prior is eliminated. Otherwise, it remains active.

**Algorithm 1** Prior Elimination GP-TS (PE-GP-TS)

---

**input** Horizon $T$, prior functions $\{\mu_{1,p}, k_{1,p}\}_{p \in P}$, confidence parameters $\{\beta_t\}_{t=1}^T$ and $\{\xi_t\}_{t=1}^T$.
1: $P_1 = P$, $S_{0,p} = \varnothing \ \forall p \in P$
2: **for** $t = 1, 2 \dots, T$ **do**
3:     Sample $\tilde{f}_{t,p} \sim \mathcal{GP}(\mu_{t,p}, k_{t,p}) \ \forall p \in P_t$
4:     Set $x_t, p_t = \arg\max_{x,p \in \mathcal{X} \times P_{t-1}} \tilde{f}_{t,p}(x)$
5:     $S_{t,p_t} = S_{t-1,p_t} \cup \{t\}$ and $S_{t,p} = S_{t-1,p}$ for $p \in P \setminus \{p_t\}$
6:     Observe $y_t = f(x_t) + \epsilon_t$
7:     Set $\eta_t = y_t - \mu_{t,p_t}(x_t)$
8:     Set $V_t = \sqrt{\xi_t|S_{t,p_t}|} + \sum_{i \in S_{t,p_t}} \sqrt{\beta_i}\sigma_{i,p_t}(x_i)$
9:     **if** $\left|\sum_{i \in S_{t,p_t}} \eta_i\right| > V_t$ **then**
10:         $P_{t+1} = P_t \setminus \{p_t\}$
11:     **else**
12:         $P_{t+1} = P_t$

---

The PE-GP-TS algorithm is presented in Algorithm 1. Similar to PE-GP-UCB, the set $S_{t,p}$ is used to store the time steps where prior $p$ was selected up to and including time $t$. When prior $p_t$ is selected, the prediction error $\eta_t = y_t - \mu_{t,p_t}(x_t)$ between the observed and predicted value made by the prior $p_t$ is computed. If the sum of prediction errors made by the prior $p_t$ exceeds the threshold value $V_t$, then $p_t$ is eliminated from the active priors $P_t$, see line 9. Note that at time step $t$, only the selected prior $p_t$ can be eliminated. As such, if a prior is very pessimistic it may never be selected and therefore will never be eliminated. Thus, the final set of active priors $P_T$ should be viewed as non-eliminated priors rather than necessarily being reasonable priors.

### 3.2 HYPERPRIOR THOMPSON SAMPLING

In our first algorithm, we removed one layer of optimism. In our second algorithm, we adopt a fully Bayesian algorithm by using a hyperposterior sampling scheme where both the prior and the mean function are sampled from their respective posteriors. By shedding the optimism over the selected prior $p_t$, HP-GP-TS should be able avoid costly exploration by selecting likely priors instead of optimistic ones.

The algorithm is visualized in Fig. 1 and presented with more details in Algorithm 2. In the first step, the current prior $p_t$ is sampled from the hyperposterior $P_{t-1}$. Then, a single sample $\tilde{f}_t$ is taken from the selected posterior $\mathcal{GP}(\mu_{t,p_t}, k_{t,p_t})$ and is used to select the current arm: $x_t = \arg\max_{x \in \mathcal{X}} \tilde{f}_t(x)$. After observing $y_t$, the hyperposterior is updated by computing the likelihood of $y_t$ under the different priors. Note that since the set of priors $P$ is finite, computing the posterior is tractable albeit computationally costly with a

---

**Algorithm 2** HyperPrior GP-TS (HP-GP-TS)

**input** Horizon $T$, prior functions $\{\mu_{1,p}, k_{1,p}\}_{p \in P}$, hyperprior $P_1$.
1: **for** $t = 1, 2 \ldots, T$ **do**
2:     Sample $p_t \sim P_t$
3:     Sample $\tilde{f}_t \sim \mathcal{GP}(\mu_{t,p_t}, k_{t,p_t})$
4:     Set $x_t = \arg\max_x \tilde{f}_t$
5:     Observe $y_t = f(x_t) + \epsilon_t$
6:     Set $P_{t+1}(p) \propto \mathbb{P}(y_t|x_t, \{x_i, y_i\}_{i=1}^{t-1}, p) \cdot P_t(p)$
      ▷ Update hyperposterior

---

complexity of $\mathcal{O}(t^3|P|)$. The likelihood $\mathbb{P}(y_t|x_t, \{x_i, y_i\}_{i=1}^{t-1}, p) = \mathcal{N}(y_t; \mu_{t,p}(x_t), \sigma_{t,p}^2(x_t) + \sigma^2)$ is simply the Gaussian likelihood of the posterior at $x_t$ with added Gaussian noise with variance $\sigma^2$.

## 4 REGRET ANALYSIS

In this section, we analyze the regret for the proposed algorithms. Recall from the problem statement that we consider two slightly different settings for the two algorithms. Specifically, for PE-GP-TS we assume the unknown prior $p^*$ is selected arbitrarily from $P$ whilst for HP-GP-TS we assume that the unknown prior $p^*$ is selected from a known hyperprior distribution $P_1$.

### 4.1 ANALYSIS OF PE-GP-TS

Ziomek et al. (2025) structured the proof of the regret bound of PE-GP-UCB into 4 larger steps; First, showing that $p^*$ is never eliminated with high probability. Second, establishing a bound on the simple regret. Third, bounding the cumulative regret. Finally, the cumulative bound is re-expressed in terms of the worst-case MIG. For PE-GP-TS, we establish a new bound on the simple regret and then adapt the steps of Ziomek et al. to accommodate the new simple regret bound.

To bound the simple regret, we require two concentration inequalities to hold for both the posteriors and the posterior samples which we present in the following lemma.

**Lemma 4.1.** *If $f(x) \sim \mathcal{GP}(\mu_{1,p^*}, k_{1,p^*})$ and $\beta_t = 2\log\left(\frac{|\mathcal{X}||P|\pi^2 t^2}{3\delta}\right)$. Then, with probability at least $1 - \delta$, the following holds for all $t, x, p \in [T] \times \mathcal{X} \times P$:*

$$|f(x) - \mu_{t,p^*}(x)| \leq \sqrt{\beta_t}\sigma_{t,p^*}(x), \tag{4}$$

$$|\tilde{f}_{t,p}(x) - \mu_{t,p}(x)| \leq \sqrt{\beta_t}\sigma_{t,p}(x). \tag{5}$$

All proofs can be found in Section A. Lemma 4.1 is based on Lemma 5.1 of Srinivas et al. (2012) but adapted to TS by specifying that it holds for any sequence of $x_1, \ldots, x_T$, as discussed by Russo & Van Roy (2014). Additionally, we add Eq. (5) which can be shown through the same steps and an additional union bound over $P$. Next, we state our bound for the simple regret of PE-GP-TS.

**Lemma 4.2.** *If the event of Lemma 4.1 holds, then the following holds for the simple regret of PE-GP-TS for all $t \in [T]$:*

$$f(x^*) - f(x_t) \leq 2\sqrt{\beta_t}\sigma_{t,p^*}(x^*) + \sqrt{\beta_t}\sigma_{t,p_t}(x_t) - \eta_t + \epsilon_t. \tag{6}$$

Compared to the simple regret bound for PE-GP-UCB, we obtain the additional term $2\sqrt{\beta_t}\sigma_{t,p^*}(x^*)$ which leads to the following regret bound:

**Theorem 4.3.** *Let $B_{p^*} = \beta_1 + \sup_{x \in \mathcal{X}} |\mu_{1,p^*}(x)|$ and $C = 2/\log(1 + \sigma^{-2})$. If $p^* \in P$ and $f \sim \mathcal{GP}(\mu_{1,p^*}, k_{1,p^*})$, then PE-GP-TS with confidence parameters $\beta_t = 2\log(2|\mathcal{X}||P|\pi^2 t^2/3\delta)$ and $\xi_t = 2\sigma^2\log(|P|\pi^2 t^2/3\delta)$, satisfies the following regret bound with probability at least $1 - \delta$:*

$$R(T) \leq 2|P|B_{p^*} + 2\sqrt{\xi_T|P|T} + 2\sqrt{CT\beta_T\hat{\gamma}_T|P|} + 2\sqrt{CT\beta_T\sum_{t \in [T]}\sigma_{t,p^*}^2(x^*)} \tag{7}$$

The bound of the first three terms is of order $\mathcal{O}(\sqrt{T\beta_T\hat{\gamma}_T})$ w.r.t. $T$ which matches that of PE-GP-UCB. To our knowledge, the best lower bound for standard GP bandits in the Bayesian setting, where $f$ is sampled from a GP, is $\Omega(\sqrt{T})$ for $d = 1$ (Scarlett, 2018). This would suggest that our bound is tight up to a factor $\mathcal{O}(\sqrt{\beta_T\hat{\gamma}_T})$ when considering only the first three terms. However, note that we have not demonstrated that $\sum_{t\in[T]}\sigma^2_{t,p^*}(x^*)$ is sublinear.

## 4.2 ANALYSIS OF HP-GP-TS

We analyze the regret of HP-GP-TS in two ways: using the UCB-based framework of Russo & Van Roy (2014) and the information-theoretic framework of Russo & Van Roy (2016). First, note that HP-GP-TS inherits the probability matching property of GP-TS that $x_t|H_t \stackrel{d}{=} x^*|H_t$ where $\stackrel{d}{=}$ denotes equal in distribution. In addition, $p_t|H_t \stackrel{d}{=} p^*|H_t$ since $p_t$ is sampled from the posterior distribution of $p^*$. In the UCB-based framework, we decompose the regret into three terms:

$$\mathbb{E}[f(x^*) - f(x_t)] = \mathbb{E}\Big[\underbrace{f(x^*) - U_{t,p^*}(x^*)}_{(1)} + \underbrace{U_{t,p^*}(x^*) - U_{t,p^*}(x_t)}_{(2)} + \underbrace{U_{t,p^*}(x_t) - f(x_t)}_{(3)}\Big] \quad (8)$$

where $U_{t,p}(x) = \mu_{t,p}(x) + \sqrt{\beta_t}\sigma_{t,p}(x)$. Summing over $t \in [T]$, the first term can be bounded by a constant whilst the third term is bounded by $\sqrt{CT\beta_T\bar{\gamma}_T}$. Together, these two terms match the Bayesian regret bound for GP-TS with known prior. For the second term, one can utilize that $p^*, x_t|H_t \stackrel{d}{=} p_t, x^*|H_t$ to re-express it as $\mathbb{E}[U_{t,p^*}(x^*) - U_{t,p_t}(x^*)]$. Hence, the second term can be seen as the cost of learning the true prior. Intuitively, if the priors are similar then the upper-confidence bounds of the true and selected prior will not differ significantly. Similarly, if the priors are sufficiently distinguishable, one would expect $P_t(p^*)$ to increase quickly from a few samples. In the following lemma, we demonstrate the latter for $|P| = 2$ with shared kernel functions.

**Lemma 4.4.** *If $|P| = 2$ and the two priors share the same kernel function ($k_p = k \; \forall p \in P$), then for any fixed sequence of arms $x_{1:t} = \{x_i\}_{i=1}^t$ the posterior probability of the true prior $p^*$ satisfies*

$$\mathbb{E}_{\mathbf{y}}\left[P_{t+1}(p)\big|p^* = p, x_{1:t}\right] \geq 1 + P_0(p)e^{\|\boldsymbol{\Sigma}^{-\frac{1}{2}}\boldsymbol{\mu}\|^2}\Phi\Big(-\frac{3}{2}\|\boldsymbol{\Sigma}^{-\frac{1}{2}}\boldsymbol{\mu}\|\Big) - \frac{1}{P_0(p)}\Phi\Big(-\frac{\|\boldsymbol{\Sigma}^{-\frac{1}{2}}\boldsymbol{\mu}\|}{2}\Big) \quad (9)$$

*where $\boldsymbol{\mu} \in \mathbb{R}^t$ such that $(\boldsymbol{\mu})_i = \mu_{p^*}(x_i) - \mu_{\tilde{p}}(x_i)$ for $\tilde{p} \neq p^*$ and $(\boldsymbol{\Sigma})_{i,j} = k(x_i, x_j)$.*

In Lemma 4.4, $\boldsymbol{\mu}$ is the difference in mean between the two priors for the given sequence of arms and $\boldsymbol{\Sigma}^{-\frac{1}{2}}\boldsymbol{\mu}$ is the whitened difference vector. As the norm of the whitened difference vector increases, $\|\boldsymbol{\Sigma}^{-\frac{1}{2}}\boldsymbol{\mu}\|$, the sum of the two rightmost terms in Eq. (9) goes to zero from below. The norm $\|\boldsymbol{\Sigma}^{-\frac{1}{2}}\boldsymbol{\mu}\|$, and consequently the posterior probability of the true prior $P_{t+1}(p^*)$, is maximized when arms with large difference in prior mean (relative to their variance) and with low correlation are selected. ~~From Lemma 4.4, we note that $P_t(p^*)$ increases the most when $\|\boldsymbol{\Sigma}^{-\frac{1}{2}}\boldsymbol{\mu}\|$ is maximized, which corresponds to selecting arms with low correlation and large difference in prior mean (relative to the variance).~~ Hence, $P_t(p^*)$ increases quickly at first when HP-GP-TS explores but levels out when the algorithm starts exploiting. Next, we state the Bayesian regret bound for HP-GP-TS.

**Theorem 4.5.** *If $p^* \sim P_1$, $f \sim \mathcal{GP}(\mu_{1,p^*}, k_{1,p^*})$ and $\beta_t = 2\log(|\mathcal{X}|t^2/\sqrt{2\pi})$, then the Bayesian regret of HP-GP-TS is bounded by*

$$BR(T) \leq \pi^2/6 + \sum_{t\in[T]}\mathbb{E}\left[U_{t,p^*}(x^*) - U_{t,p_t}(x^*)\right] + \sqrt{CT\beta_T\bar{\gamma}_T}. \quad (10)$$

Unlike PE-GP-TS and PE-GP-UCB, the regret bound of the third term for HP-GP-TS depends on the average MIG $\sqrt{\bar{\gamma}_T}$ rather than the worst case $\sqrt{|P|\hat{\gamma}_T}$ which can impact the theoretical regret significantly if the complexity of the priors differ and the prior is weighted towards simple priors. This is reasonable since the elimination methods assume arbitrary selection of $p^*$ as opposed to sampling from a hyperprior. If the hyperprior is deterministic then the regret bound for HP-GP-TS matches that of GP-TS up to a factor $\mathcal{O}(\sqrt{\log T})$ (Takeno et al., 2024) and $\bar{\gamma}_T$ would be equal to the worst case $\hat{\gamma}_T$. Again, using the lower bound of Scarlett (2018), our upper bound would be tight up to a factor of $\mathcal{O}(\sqrt{\beta_T\bar{\gamma}_T})$ when considering only the first and third terms. We have not shown that

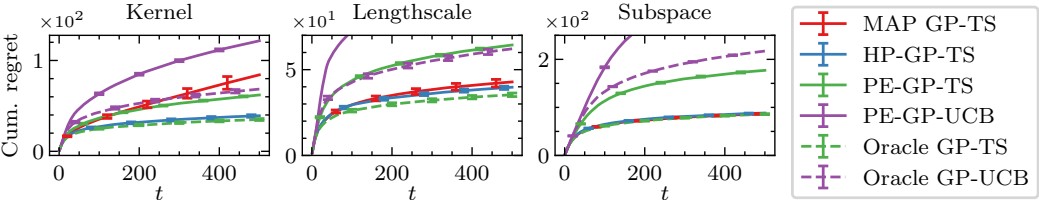

Figure 2: Cumulative regret for synthetic experiments with varying kernel (left), lengthscale (center) and mean function (right). The average final regret for PE-GP-UCB is 116.5 and 389.0 in the lengthscale and subspace experiments. Errorbars correspond to $\pm 1$ standard error.

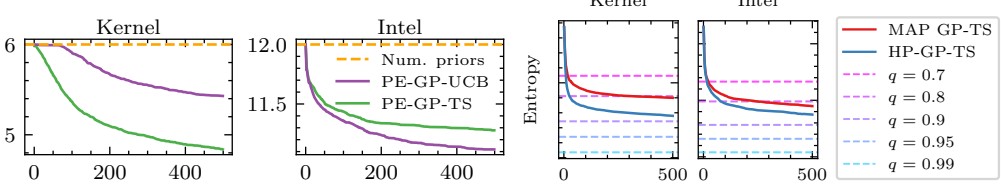

Figure 3: Mean number of priors remaining in $P_t$ over time for PE-GP-UCB and -TS (left). Mean entropy in the hyperposterior $P_t$ over time for HP- and MAP GP-TS (right). The dashed reference values correspond to entropies of discrete distributions with prob. $q$ on one choice and prob. $\frac{1-q}{|P|-1}$ on the other $|P|-1$ choices.

$\sum_{t\in[T]}\mathbb{E}[U_{t,p^*}(x^*) - U_{t,p_t}(x^*)]$ is sublinear. As noted before, this sum corresponds to the cost of learning the true prior. Conceptually, we expect the terms in the sum to decrease over time as either $p_t$ or $x_t$ converge to $p^*$ or $x^*$ respectively. Empirically, we observe this in our synthetic experiments, see Fig. 10 in Section D

The information-theoretic framework of Russo & Van Roy (2016) can be applied generally if the probability matching property is satisfied and the rewards are subgaussian (Vashishtha & Maillard, 2025). This framework ignores the structural assumptions imposed by sampling $f$ from a Gaussian process and therefore provides weak guarantees. However, we invoke it to demonstrate sublinearity when the number of arms $|\mathcal{X}|$ is smaller than the horizon $T$.

**Theorem 4.6.** *If $p^* \sim P_1$, $f \sim \mathcal{GP}(\mu_{1,p^*}, k_{1,p^*})$, the Bayesian regret of HP-GP-TS is bounded by*

$$BR(T) \leq \sqrt{2|\mathcal{X}|\log(|\mathcal{X}|)(\sigma_0^2 + \sigma^2)T}. \tag{11}$$

The proof of Theorem 4.6 follows the proof in section D.2 of Russo & Van Roy (2016) with subgaussian noise. For completeness, we provide a proof using the GP-bandit notation in Algorithm 2. The information-theoretic regret bound is $\mathcal{O}(\sqrt{T})$ which improves upon the $\mathcal{O}(\sqrt{T\beta_T\bar{\gamma}_T})$ obtained previously and would match the lower bound of Scarlett (2018) in terms of $T$. However, we also obtain a $\mathcal{O}(\sqrt{|\mathcal{X}|\log|\mathcal{X}|})$ dependency. Note that this bound is only sublinear when $|\mathcal{X}|\log|\mathcal{X}| < T$.

## 5 EXPERIMENTS

**Synthetic experiments** We consider three synthetic setups with different choices of priors in $P$. For the first setup, the priors have one of the following kernels: i) RBF kernel, ii) the rational quadratic kernel with $\alpha = 0.5$, iii) Matérn kernel with $\nu = 5/2$, iv) Matérn kernel with $\nu = 3/2$, v) periodic kernel with period $\rho = 5$, vi) linear kernel with $v = 0.05^2$. All kernels use a lengthscale of $1.0$ and are scaled s.t. $k(x,\tilde{x}) \leq 1$. In addition, the mean function for all priors is zero everywhere. For the second setup, the priors use the RBF kernel with lengthscales $4$, $2$, $1$ or $1/2$. For the third setup, the total dimensions $d = 16$ but each prior $p_i$ assumes $f(x)$ depends on $d_s = 4$ subdimensions: $[i, i+1, i+2, i+3]$ for $i \in [5]$. Dimensions larger than 5 are wrapped around 1, i.e. $((j-1) \mod 5) + 1$, such that the priors are equally difficult to distinguish and optimize. All priors use the RBF kernel with lengthscale $\ell = 8$. For all three setups, the true prior $p^*$ is sampled uniformly from

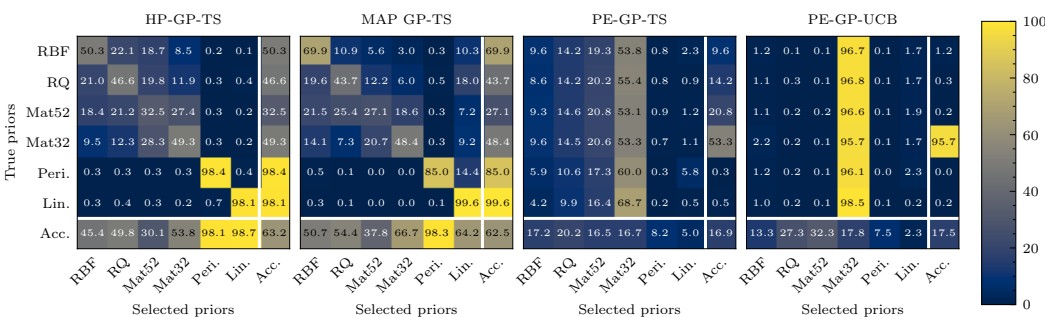

Figure 4: Confusion matrices for the true prior $p^*$ and the selected priors $p_t$ for the kernel experiment. Row-wise normalized to 100%.

$P$, the noise variance $\sigma^2 = 0.25^2$, and the horizon $T = 500$. For the first two setups, 500 arms are equidistantly spaced in $[0, 20]$ and for the third 500 arms are sampled uniformly on $[0, 20]^{16}$. The prior elimination methods use $\delta = 0.05$. All models are evaluated on 500 seeds on each setup. As baselines, we use PE-GP-UCB and Maximum A Posteriori (MAP) GP-TS where MAP GP-TS is identical to HP-GP-TS except for greedily selecting $p_t$ from the posterior: $p_t = \arg\max_p P_{t-1}(p)$[1]. Regardless of the selected prior, the weighting of the hyperposterior $P_t$ is updated for all priors. Hence, greedily selecting the prior could reduce unnecessary exploration. In addition, we investigate the oracle variants of PE-GP-TS and PE-GP-UCB with $\delta = 0.05$ that are only given the true prior: $P_1 = \{p^*\}$.

The cumulative regret for the three synthetic experiments is shown in Fig. 2. Across all three experiments, we observe that HP-GP-TS has lower regret than the other methods and performs close to the oracle GP-TS. For the kernel and subspace experiments, PE-GP-TS has lower regret than the oracle GP-UCB. Hence, even if PE-GP-UCB was optimized to perform as well as the oracle, it would still not achieve the regret of our proposed methods. MAP GP-TS has slightly higher regret than HP-GP-TS for the lengthscale and subspace experiments but has significantly higher regret and variance for the kernel experiment. The greedy selection of MAP (MLE) leads to under-exploration for MAP GP-TS in certain instances.

The number of priors remaining $|P_t|$ and the hyperposterior entropy for the kernel experiment is shown in Fig. 3. The PE-methods eliminate at most one prior on average. In contrast, the hyperposterior entropy of HP-GP-TS is equivalent to 80-90% of the probability mass being assigned to one prior. HP- and MAP-GP-TS thus effectively discards priors at a much faster rate. The same pattern holds for the lengthscale and subspace experiments, see Figs. 8 and 9 in Section D.

In Fig. 4, we visualize how often the methods select the true prior $p^*$ (or kernel) in the kernel experiment as confusion matrices. PE-GP-UCB selects the Matérn-3/2 kernel more than 96% of the rounds. The Matérn-3/2 kernel induces a distribution over functions that are less smooth compared to the other kernels and produces much higher confidence intervals outside the observed data leading to excessive optimistic exploration. PE-GP-TS also shows a bias towards the Matérn-3/2 kernel but does not select it as frequently as PE-GP-UCB - demonstrating that one layer of optimism has been removed. The overall "accuracy" of the selected priors, i.e. $\sum_{t \in [T]} \mathbb{1}\{p_t = p^*\}/T$, for the elimination-based methods is around 17% in the kernel experiment compared to 62.5% and 63.2% for MAP and HP-GP-TS respectively. For HP-GP-TS, we observe that it can easily identify the periodic and linear kernels. However, the RBF, Matérn and RQ kernels are often confused with each other. These kernels do not have as easily distinguishable characteristics and are likely to produce similar posteriors even with a small amount of data. See Fig. 11 in Section D for confusion matrices in the lengthscale and subspace experiments.

**Scaling $|P|$** We perform two experiments to understand how the regret of our algorithms scale with the number of priors. In both experiments, the average difficulty of the problem is kept constant such that the regret of the oracle models is constant. In the first experiment, we increase the discretization of the lengthscale values. The lengthscales are equidistantly spaced in $[0.5, 4]$ with

---

[1]Note that since the hyperprior is uniform, MAP is equivalent to discrete maximum likelihood estimation.

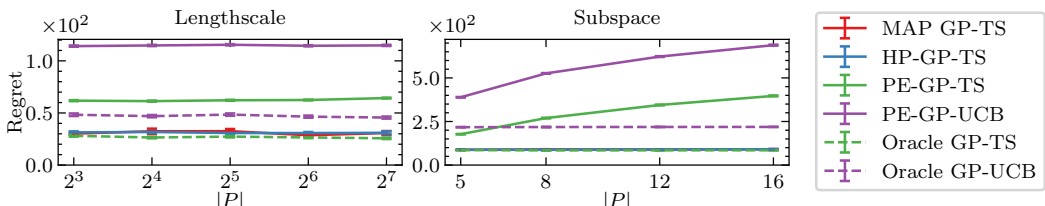

Figure 5: Total regret for the lengthscale and subspace experiments as $|P|$ increases.

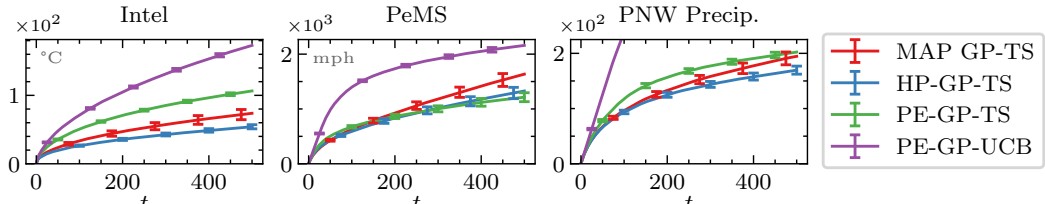

Figure 6: Cumulative regret on Intel temperature data (top) and PeMS data (bottom). Errorbars correspond to $\pm 1$ standard error. The average final regret for PE-GP-UCB is 511.9 for PNW.

$|P| \in \{8, 16, 32, 64, 128\}$. As $|P|$ increases, the difference between similar priors is reduced. In the second experiment, we increase the number of priors in the subspace experiment from 5 up to 16. Each prior can share at most 3 out of 4 dimensions with other priors which ensures the priors remain meaningfully different. The total regret as the number of priors increases is shown in Fig. 5. For the lengthscale experiment, increasing the number of priors above 8 does not affect the regret for any algorithm. This is likely due to the redundancy in the priors. However in the subspace experiment, the regret of the prior elimination algorithms scales approximately as $\sqrt{|P|}$ whilst MAP- and HP-GP-TS are consistently close to the constant regret of the oracle.

**Real-world data** We perform three experiments with real-world data from the Intel Berkeley dataset (Madden et al., 2004), California Performance Measurement System (PeMS) (Chen et al., 2001; California Department of Transportation, 2024) and Pacific Northwest (PNW) daily precipitation dataset (Widmann & Bretherton, 1999; 2000). Each dataset contains measurements from a set of sensors over time. We split each dataset into a training and test set where the test set contains the last third of the data. Hence, the distribution of the test data may have shifted slightly from the training data. The training sets are split further into separate buckets to define our priors. For each bucket $p$, we compute the empirical mean $\hat{\mu}_p$ and covariance $\hat{\Sigma}_p$ which defines the prior $\mathcal{GP}(\hat{\mu}_p, \hat{\Sigma}_p)$. The buckets in the Intel data corresponds to the 12 days in the training dataset. For the PeMS data, each hour between 06:00 and 13:00 defines one prior, giving 7 priors. For the daily precipication data, each month in the year constitutes a prior, yielding 12 priors. When running the experiments, we select a measurement of all sensors from the test data uniformly at random. The selected measurements correspond to the unknown function $f(x)$ where $x$ is the sensor index and the goal is then to identify sensors measuring large temperatures, small speeds or high precipication respectively for the three datasets. When the algorithms select an arm to evaluate, we add Gaussian noise to $y_t$ with variance $\sigma^2$ around 5% of the signal variance, similar to Srinivas et al. (2012); Bogunovic et al. (2016). See Section C for more details about the experimental setup.

The cumulative regret for the experiments with real-world data is presented in Fig. 6. For the Intel and PNW data, HP- and MAP GP-TS obtain the lowest and second lowest cumulative regret respectively. HP- and MAP GP-TS have lower regret than PE-GP-TS initially but PE-GP-TS catches up and has the lowest total regret. To understand this better, we visualize quantiles of the total regret in Fig. 13. MAP- and HP-GP-TS have the lowest median regret for all three experiments and hence perform best in a majority of instances. However, the 90th and 95th quantiles are considerably larger for the PeMS data which impacts the average regret significantly. Hence, for the PeMS data, the prior elimination methods seem to yield more stable results.

In Fig. 3, the number of priors remaining in $|P_t|$ and the hyperposterior entropy is shown for the Intel experiment. Similar to the synthetic experiment, on average, the prior elimination methods eliminate

less than 1 prior whereas the hyperposteriors of HP- and MAP GP-TS concentrate the equivalent of 80-90% of the probability mass to one prior. The results for PeMS and PNW experiment are shown in Figs. 8 and 9 in Section D. Here, effectively no priors are eliminated. For the PNW experiment, the hyperposteriors do not concentrate as much compared to the other experiments. This could indicate that knowing the exact prior is not as important for the PNW data.

# 6 CONCLUSION

In this paper, we have proposed two algorithms for joint prior selection and regret minimization in GP bandits based on GP-TS. We have analyzed the algorithms theoretically and have experimentally evaluated both algorithms on synthetic and real-world data. We find that they both select the true prior more often and obtain lower regret than previous work due to lowering the amount of optimistic exploration.

## ETHICS STATEMENT

The work of this paper introduces general algorithms for efficient blackbox optimization when parameters of the blackbox cannot be uniquely determined beforehand. Overall, blackbox optimization is used in many different disciplines where the typical goal is to improve allocation of resources or increase resource output. We believe such applications are beneficial to society at large and believe our work does not raise any ethical concerns. The datasets used in this paper do not involve any human or animal subjects.

## REPRODUCIBILITY STATEMENT

The proposed algorithms are described in Section 3 with more details provided in Algorithms 1 and 2. For the theoretical results, we describe the overall assumptions of the problem in Section 2 and provide more detailed assumptions in the respective theorems and lemmas in Section 4. We provide detailed proofs of all theorems and lemmas in Section A. The experimental setup is described in Section 5 with further details about the data processing and results provided in Sections C and D. All experimental results are averaged across 500 runs with fixed seeds to ensure reproducibility. A link to an anonymous repository containing the source code will be posted as a comment on OpenReview.

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

# A    PROOFS

In the following section, we state and prove the results shown in the main text.

## A.1    PE-GP-TS

First, we state and prove concentration inequalities for $f(x)$ and $\tilde{f}_{t,p}(x)$.

**Lemma 4.1.** *If $f(x) \sim \mathcal{GP}(\mu_{1,p^*}, k_{1,p^*})$ and $\beta_t = 2\log\left(\frac{|\mathcal{X}||P|\pi^2 t^2}{3\delta}\right)$. Then, with probability at least $1 - \delta$, the following holds for all $t, x, p \in [T] \times \mathcal{X} \times P$:*

$$|f(x) - \mu_{t,p^*}(x)| \leq \sqrt{\beta_t}\sigma_{t,p^*}(x), \tag{4}$$

$$|\tilde{f}_{t,p}(x) - \mu_{t,p}(x)| \leq \sqrt{\beta_t}\sigma_{t,p}(x). \tag{5}$$

*Proof.* Follows by the same steps as Lemma 5.1 of Srinivas except we condition on the complete history $H_t$ instead of only $\mathbf{y}_{1:t-1}$. Additionally, for Eq. (5) we must take an additional union bound over $p \in P$.

Fix $t, x, p \in [T] \times \mathcal{X} \times P$. Given the history $H_t$, $\tilde{f}_{t,p}(x) \sim \mathcal{N}(\mu_{t,p}(x), \sigma_{t,p}^2(x))$. Using that $\mathbb{P}(Z > c) \leq 1/2 e^{-c^2/2}$ for $Z \sim \mathcal{N}(0,1)$, we get that

$$\mathbb{P}\left(\left|\frac{\tilde{f}_{t,p}(x) - \mu_{t,p^*}(x)}{\sigma_{t,p^*}(x)}\right| > \sqrt{\beta_t}\right) \leq \exp(-\beta_t/2) \tag{12}$$

$$= \frac{3\delta}{|\mathcal{X}||P|\pi^2 t^2} \tag{13}$$

Note that $\sum_{t \geq 1} \frac{1}{t^2} = \frac{\pi^2}{6}$ By taking the union bound over $\mathcal{X}$, $P$ and $t \geq 1$, Eq. (5) holds w.p. at least $1 - \delta/2$. By the same reasoning and skipping the union bound over $P$, Eq. (4) holds w.p. at least $1 - \delta/2$. Thus, both events hold w.p. at least $1 - \delta$. □

Next, we state three lemmas from Ziomek et al. (2025) that are used in the proof of our regret bound.

**Lemma A.1.** *(Lemma 5.1 of Ziomek et al. (2025))* *If $\xi_t = 2\sigma^2 \log\left(\frac{|P|\pi^2 t^2}{6\delta}\right)$, then the following holds with probability at least $1 - \delta$:*

$$\left|\sum_{i \in S_{t,p}} \epsilon_i\right| \leq \sqrt{\xi_t |S_{t,p}|} \quad \forall t, p \in [T] \times P. \tag{14}$$

**Lemma A.2.** *(Lemma 5.2 of Ziomek et al. (2025))* *Let $B_{p^*} = \beta_1 + \sup_{x \in \mathcal{X}} |\mu_{1,p^*}(x)|$, then if $\mu_{1,p^*}$ and $k_{1,p^*}$ satisfy $|\mu_{1,p^*}(\cdot)| < \infty$ and $k_{1,p^*}(\cdot, \cdot) \leq 1$ and Lemma 4.1 holds, then*

$$\sup_{x \in \mathcal{X}} |f(x)| \leq B_{p^*}. \tag{15}$$

**Lemma A.3.** *(Lemma 5.3 of Ziomek et al. (2025))* *For $C = 2/\log(1 + \sigma^{-2})$, $\sum_{t \notin \mathcal{C}} \sqrt{\beta_t}\sigma_{t,p_t}(x_t) \leq \sqrt{CT\beta_T\hat{\gamma}_T |P|}$ where $\beta_T = \max_{p \in P} \beta_T$ and $\hat{\gamma}_T = \max_{p \in P} \gamma_{T,p}$.*

Then, we state and prove the new simple regret bound for PE-GP-TS.

**Lemma 4.2.** *If the event of Lemma 4.1 holds, then the following holds for the simple regret of PE-GP-TS for all $t \in [T]$:*

$$f(x^*) - f(x_t) \leq 2\sqrt{\beta_t}\sigma_{t,p^*}(x^*) + \sqrt{\beta_t}\sigma_{t,p_t}(x_t) - \eta_t + \epsilon_t. \tag{6}$$

*Proof.* First, we upper bound $f(x^*)$ as follows

$$f(x^*) \leq \mu_{t,p^*}(x^*) + \sqrt{\beta_t}\sigma_{t,p^*}(x^*) \qquad \text{(Eq. (4))} \tag{16}$$

$$\leq \tilde{f}_{t,p^*}(x^*) + 2\sqrt{\beta_t}\sigma_{t,p^*}(x^*) \qquad \text{(Eq. (5))} \tag{17}$$

$$\leq \tilde{f}_{t,p_t}(x_t) + 2\sqrt{\beta_t}\sigma_{t,p^*}(x^*). \qquad \text{(TS selection rule)} \tag{18}$$

Then, we lower bound $f(x_t)$

$$f(x_t) = \mu_{t,p_t}(x_t) + \eta_t - \epsilon_t \qquad \text{(Def. of } \eta_t) \qquad (19)$$

$$\geq \tilde{f}_{t,p_t}(x_t) - \sqrt{\beta_t}\sigma_{t,p_t}(x_t) + \eta_t - \epsilon_t. \qquad \text{(Eq. (5))} \qquad (20)$$

Combining, Eqs. (18) and (20) we obtain

$$f(x^*) - f(x_t) \leq 2\sqrt{\beta_t}\sigma_{t,p^*}(x^*) + \sqrt{\beta_t}\sigma_{t,p_t}(x_t) - \eta_t + \epsilon_t. \qquad (21)$$

$\square$

Finally, we state and prove the cumulative regret bound for PE-GP-TS.

**Theorem 4.3.** *Let $B_{p^*} = \beta_1 + \sup_{x \in \mathcal{X}} |\mu_{1,p^*}(x)|$ and $C = 2/\log(1 + \sigma^{-2})$. If $p^* \in P$ and $f \sim \mathcal{GP}(\mu_{1,p^*}, k_{1,p^*})$, then PE-GP-TS with confidence parameters $\beta_t = 2\log(2|\mathcal{X}||P|\pi^2 t^2/3\delta)$ and $\xi_t = 2\sigma^2 \log(|P|\pi^2 t^2/3\delta)$, satisfies the following regret bound with probability at least $1 - \delta$:*

$$R(T) \leq 2|P|B_{p^*} + 2\sqrt{\xi_T|P|T} + 2\sqrt{CT\beta_T\hat{\gamma}_T|P|} + 2\sqrt{CT\beta_T \sum_{t \in [T]} \sigma_{t,p^*}^2(x^*)} \qquad (7)$$

*Proof.* First, we show that the true prior $p^*$ is never rejected if Lemmas A.1 and 4.1 hold.

$$\left| \sum_{i \in S_{t,p^*}} \eta_i \right| = \left| \sum_{i \in S_{t,p^*}} (y_i - f(x_i) + f(x_i) - \mu_{i,p^*}(x_i)) \right| \qquad (22)$$

$$\leq \left| \sum_{i \in S_{t,p^*}} \epsilon_i \right| + \sum_{i \in S_{t,p^*}} |f(x_i) - \mu_{i,p^*}(x_i)| \qquad \text{(Triangle ineq.)} \qquad (23)$$

$$\leq \sqrt{\xi_t|S_{t,p^*}|} + \sum_{i \in S_{t,p^*}} \sqrt{\beta_i}\sigma_{i,p^*}(x_i). \qquad \text{(Lemmas A.1 and 4.1)} \qquad (24)$$

Next, we bound the cumulative regret. To establish a bound on the cumulative regret, we must separate out the rounds where priors are eliminated. Hence, define the set of critical iterations as

$$\mathcal{C} = \left\{ t \in [T] : \left| \sum_{i \in S_{t,p_t}} \eta_i \right| > \sqrt{\xi_t S_{t,p_t}} + \sum_{i \in S_{t,p_t}} \sqrt{\beta_i}\sigma_{i,p_t}(x_i) \right\}. \qquad (25)$$

Using Lemma A.2 and Eq. (21), we can bound the cumulative regret as follows:

$$\text{BR}(T) = \sum_{t \in \mathcal{C}} \text{BR}_t + \sum_{t \notin \mathcal{C}} \text{BR}_t \qquad (26)$$

$$\leq 2|P|B_{p^*} + \sum_{t \notin \mathcal{C}} 2\sqrt{\beta_t}\sigma_{t,p^*}(x^*) + \sum_{t \notin \mathcal{C}} \sqrt{\beta_t}\sigma_{t,p_t}(x_t) + \sum_{p \in P} \sum_{t \in S_{T,p} \backslash \mathcal{C}} (\epsilon_t - \eta_t). \qquad (27)$$

where $B_{p^*} := \beta_1 + \sup_{x \in \mathcal{X}} |\mu_{1,p^*}(x)|$. If $t \notin \mathcal{C}$, line 9 in Algorithm 1 evaluates to `false` and hence

$$\sum_{p \in P} \sum_{t \in S_{T,p} \backslash \mathcal{C}} -\eta_t \leq \sum_{p \in P} \sqrt{\xi_T|S_{T,p}|} + \sum_{p \in P} \sum_{t \in S_{T,p} \backslash \mathcal{C}} \sqrt{\beta_t}\sigma_{t,p}(x_t). \qquad (28)$$

Additionally, using Lemma A.1, we can bound the Gaussian noise:

$$\sum_{p \in P} \sum_{t \in S_{T,p} \backslash \mathcal{C}} \epsilon_t \leq \sum_{p \in P} \left| \sum_{t \in S_{T,p} \backslash \mathcal{C}} \epsilon_t \right| \leq \sum_{p \in P} \left| \sum_{t \in S_{T,p}} \epsilon_t \right| \qquad (29)$$

$$\leq \sum_{p \in P} \sqrt{\xi_T|S_{T,p}|} \qquad \text{(Lemma A.1)} \qquad (30)$$

$$\leq \sqrt{\xi_T|P|T} \qquad \text{(Cauchy-Schwarz)} \qquad (31)$$

Combining the above, the cumulative regret is bounded by

$$\text{BR}(T) \leq 2|P|B_{p^*} + 2\sqrt{\xi_T|P|T} + 2\sum_{t \notin \mathcal{C}} \sqrt{\beta_t}\sigma_{t,p^*}(x^*) + 2\sum_{t \notin \mathcal{C}} \sqrt{\beta_t}\sigma_{t,p_t}(x_t). \tag{32}$$

Finally, applying Lemma A.3, we obtain the result

$$\text{BR}(T) \leq 2|P|B_{p^*} + 2\sqrt{\xi_T|P|T} + 2\sqrt{CT\beta_T \sum_{t \in [T]} \sigma_{t,p^*}^2(x^*)} + 2\sqrt{CT\beta_T\hat{\gamma}_T|P|}. \tag{33}$$

$\qquad\qquad\qquad\qquad\qquad\qquad\qquad\qquad\qquad\qquad\qquad\qquad\qquad\qquad\qquad\qquad\square$

## A.2 UCB-ANALYSIS OF HP-GP-TS

Next, we state and prove our UCB-based regret bound for HP-GP-TS.

**Theorem 4.5.** *If $p^* \sim P_1$, $f \sim \mathcal{GP}(\mu_{1,p^*}, k_{1,p^*})$ and $\beta_t = 2\log(|\mathcal{X}|t^2/\sqrt{2\pi})$, then the Bayesian regret of HP-GP-TS is bounded by*

$$BR(T) \leq \pi^2/6 + \sum_{t \in [T]} \mathbb{E}\left[U_{t,p^*}(x^*) - U_{t,p_t}(x^*)\right] + \sqrt{CT\beta_T\bar{\gamma}_T}. \tag{10}$$

*Proof.* To begin, we note that $x_t|H_t \overset{d}{=} x^*|H_t$ and $p_t|H_t \overset{d}{=} p^*|H_t$ since both $x_t$ and $p_t$ are sampled from their respective posteriors. Let $U_{t,p}(x) := \mu_{t,p}(x) + \sqrt{\beta_t}\sigma_{t,p}(x)$. Then, we start decomposing the instant regret into two terms:

$$\text{BR}(T) = \sum_{t \in [T]} \mathbb{E}\left[f(x^*) - f(x_t)\right] \tag{34}$$

$$= \sum_{t \in [T]} \mathbb{E}[f(x^*) - U_{t,p^*}(x^*) + U_{t,p^*}(x^*) \tag{35}$$

$$\qquad\qquad - U_{t,p_t}(x^*) + U_{t,p^*}(x_t) - f(x_t)] \qquad\qquad (x^*, p_t|H_t \overset{d}{=} x_t, p^*|H_t) \tag{36}$$

$$= \underbrace{\sum_{t \in [T]} \mathbb{E}\left[f(x^*) - U_{t,p^*}(x^*)\right]}_{(1)} + \underbrace{\sum_{t \in [T]} \mathbb{E}\left[U_{t,p_t}(x_t) - U_{t,p^*}(x_t)\right]}_{(2)} + \underbrace{\sum_{t \in [T]} \mathbb{E}\left[U_{t,p^*}(x_t) - f(x_t)\right]}_{(3)} \tag{37}$$

We begin by bounding term (1),

$$(1) = \sum_{t \in [T]} \mathbb{E}\left[f(x^*) - \mu_{t,p^*}(x^*) - \sqrt{\beta_t}\sigma_{t,p^*}(x^*)\right] \tag{38}$$

$$\leq \sum_{t \in [T]} \mathbb{E}\left[\left[f(x^*) - \mu_{t,p^*}(x^*) - \sqrt{\beta_t}\sigma_{t,p^*}(x^*)\right]_+\right] \qquad ([\cdot]_+ := \max(\cdot, 0)) \tag{39}$$

$$\leq \sum_{t \in [T]} \sum_{x \in \mathcal{X}} \mathbb{E}\left[\left[f(x) - \mu_{t,p^*}(x) - \sqrt{\beta_t}\sigma_{t,p^*}(x)\right]_+\right] \qquad (x^* \in \mathcal{X}, [\cdot]_+ \geq 0) \tag{40}$$

$$\leq \sum_{t \in [T]} \sum_{x \in \mathcal{X}} \mathbb{E}_{p^*, H_t}\left[\mathbb{E}_t\left[\left[f(x) - \mu_{t,p^*}(x) - \sqrt{\beta_t}\sigma_{t,p^*}(x)\right]_+ \big| p^*, H_t\right]\right] \qquad \text{(Tower rule)} \tag{41}$$

Recall that for $Z \sim \mathcal{N}(\mu, \sigma)$ with $\mu \leq 0$, $\mathbb{E}[[Z]_+] = \frac{\sigma}{\sqrt{2\pi}}\exp\left(\frac{-\mu^2}{2\sigma^2}\right)$. In our case, note that $f(x)|p^*, H_t \sim \mathcal{N}(\mu_{t,p^*}(x), \sigma_{t,p^*}^2(x))$ and $-\mu_{t,p^*}(x) - \sqrt{\beta_t}\sigma_{t,p^*}(x)$ is deterministic given $p^*, H_t$.

Hence,

$$(1) \leq \sum_{t \in [T]} \sum_{x \in \mathcal{X}} \mathbb{E}_{p^*, H_t} \left[ \frac{\sigma_{t,p^*}(x)}{\sqrt{2\pi}} \exp\left( \frac{-\beta_t}{2} \right) \right] \tag{42}$$

$$\leq \sum_{t \in [T]} \sum_{x \in \mathcal{X}} \mathbb{E}_{p^*, H_t} \left[ \frac{1}{\sqrt{2\pi}} \exp\left( -\frac{\beta_t}{2} \right) \right] \qquad (\sigma_{t,p^*}(x) \leq \sigma_{0,p^*}(x) \leq 1) \tag{43}$$

$$= \sum_{t \in [T]} \sum_{x \in \mathcal{X}} \frac{1}{\sqrt{2\pi}} \exp(-\beta_t/2) \tag{44}$$

$$\leq \sum_{t \in [T]} \frac{1}{t^2} \leq \frac{\pi^2}{6}. \qquad (\beta_t = 2 \log(|\mathcal{X}| t^2 / \sqrt{2\pi})) \tag{45}$$

Next, we bound (3) as follows:

$$(3) = \sum_{t \in [T]} \mathbb{E}\left[ U_{t,p^*}(x_t) - f(x_t) \right] \tag{46}$$

$$= \sum_{t \in [T]} \mathbb{E}_{H_t} \left[ \mathbb{E}_t \left[ U_{t,p^*}(x_t) - f(x_t) | H_t \right] \right] \qquad \text{(Tower rule)} \tag{47}$$

$$= \sum_{t \in [T]} \mathbb{E}_{H_t} \left[ \mathbb{E}_t \left[ U_{t,p^*}(x_t) - \mu_{t,p^*}(x_t) | H_t \right] \right] \qquad (\mathbb{E}[f(x_t)|H_t] = \mathbb{E}[\mu_{t,p^*}(x_t)|H_t]) \tag{48}$$

$$= \sum_{t \in [T]} \mathbb{E}_{H_t} \left[ \mathbb{E}_t [\sqrt{\beta_t} \sigma_{t,p^*}(x_t) | H_t] \right] \qquad (U_{t,p^*}(\cdot) = \mu_{t,p^*}(\cdot) + \sqrt{\beta_t} \sigma_{t,p^*}(\cdot)) \tag{49}$$

$$= \sum_{t \in [T]} \mathbb{E}[\sqrt{\beta_t} \sigma_{t,p^*}(x_t)] \tag{50}$$

Continuing,

$$(3) = \mathbb{E}\left[ \sum_{t \in [T]} \sqrt{\beta_t} \sigma_{t,p^*}(x_t) \right] \tag{51}$$

$$\leq \mathbb{E}\left[ \sqrt{\sum_{t \in [T]} \beta_t \sum_{t \in [T]} \sigma_{t,p^*}^2(x_t)} \right] \qquad \text{(Cauchy-Schwarz)} \tag{52}$$

$$= \sqrt{\sum_{t \in [T]} \beta_t} \, \mathbb{E}\left[ \sqrt{\sum_{t \in [T]} \sigma_{t,p^*}^2(x_t)} \right] \qquad (\beta_t \text{ deterministic}) \tag{53}$$

$$\leq \sqrt{\sum_{t \in [T]} \beta_t} \sqrt{\mathbb{E}\left[ \sum_{t \in [T]} \sigma_{t,p^*}^2(x_t) \right]} \qquad \text{(Jensen's inequality)} \tag{54}$$

$$\leq \sqrt{\beta_T T} \sqrt{\mathbb{E}_{p^*} \left[ \mathbb{E}\left[ \sum_{t \in [T]} \sigma_{t,p^*}^2(x_t) \Big| p^* \right] \right]} \qquad (\beta_t \text{ increasing}) \tag{55}$$

$$\leq \sqrt{\beta_T T} \sqrt{C \mathbb{E}_{p^*} \left[ \gamma_{T,p^*} \right]} \qquad \text{(Lemma 5.4 of Srinivas et al. (2012))} \tag{56}$$

$$\leq \sqrt{\beta_T T} \sqrt{C \bar{\gamma}_T} \tag{57}$$

Combining the bounds for (1) and (3), we obtain the desired result

$$\text{BR}(T) \leq \frac{\pi^2}{6} + \sum_{t \in [T]} \mathbb{E}\left[ U_{t,p_t}(x_t) - U_{t,p^*}(x_t) \right] + \sqrt{CT \beta_T \bar{\gamma}_T}. \tag{58}$$

$\square$

## A.3 ANALYSIS OF EXPECTED POSTERIOR PROBABILITIES

**Lemma 4.4.** *If $|P| = 2$ and the two priors share the same kernel function ($k_p = k \; \forall p \in P$), then for any fixed sequence of arms $x_{1:t} = \{x_i\}_{i=1}^t$ the posterior probability of the true prior $p^*$ satisfies*

$$\mathbb{E}_{\mathbf{y}}\left[P_{t+1}(p)\big|p^* = p, x_{1:t}\right] \geq 1 + P_0(p)e^{\|\mathbf{\Sigma}^{-\frac{1}{2}}\boldsymbol{\mu}\|^2}\Phi\left(-\frac{3}{2}\|\mathbf{\Sigma}^{-\frac{1}{2}}\boldsymbol{\mu}\|\right) - \frac{1}{P_0(p)}\Phi\left(-\frac{\|\mathbf{\Sigma}^{-\frac{1}{2}}\boldsymbol{\mu}\|}{2}\right) \quad (9)$$

*where $\boldsymbol{\mu} \in \mathbb{R}^t$ such that $(\boldsymbol{\mu})_i = \mu_{p^*}(x_i) - \mu_{\tilde{p}}(x_i)$ for $\tilde{p} \neq p^*$ and $(\mathbf{\Sigma})_{i,j} = k(x_i, x_j)$.*

*Proof.* Fix $p^* = p$ and let $\tilde{p}$ denote the incorrect prior. Without loss of generality, assume $\mu_{1,p}(x) = 0$ $\forall x \in \mathcal{X}$ such that the two priors are: $\mathcal{GP}_p(0, k)$ and $\mathcal{GP}_{\tilde{p}}(\mu, k)$. For the observed rewards $\mathbf{y} \in \mathbb{R}^t$, we use $N_p(\mathbf{y}) = \mathcal{N}(\mathbf{y}; \mathbf{0}, \mathbf{K} + \sigma^2 I)$ and $N_{\tilde{p}}(\mathbf{y}) = \mathcal{N}(\mathbf{y}; \boldsymbol{\mu}(x_{1:t}), \mathbf{K} + \sigma^2 I)$ to denote the likelihood (or multivariate Gaussian density) under the priors $p$ and $\tilde{p}$ where $\boldsymbol{\mu}(x_{1:t}) \in \mathbb{R}^t$ and $\mathbf{K} \in \mathbb{R}^{t \times t}$ s.t. $(\boldsymbol{\mu})_i = \mu(x_i)$ and $(\mathbf{K})_{i,j} = k(x_i, x_j)$. To simplify the notation, let $\boldsymbol{\mu} := \boldsymbol{\mu}(x_{1:t})$ and $\mathbf{\Sigma} = K + \sigma^2 I$ s.t. $N_p(\mathbf{y}) = \mathcal{N}(\mathbf{y}; \mathbf{0}, \mathbf{\Sigma})$ and $N_{\tilde{p}}(\mathbf{y}) = \mathcal{N}(\mathbf{y}; \boldsymbol{\mu}, \mathbf{\Sigma})$.

By Bayes' theorem, the posterior probability of $p$ satisfies:

$$P_{t+1}(p) = \frac{N_p(\mathbf{y})P_0(p)}{N_p(\mathbf{y})P_0(p) + N_{\tilde{p}}(\mathbf{y})P_0(\tilde{p})} = \frac{\frac{N_p(\mathbf{y})}{N_{\tilde{p}}(\mathbf{y})}P_0(p)}{\frac{N_p(\mathbf{y})}{N_{\tilde{p}}(\mathbf{y})}P_0(p) + (1 - P_0(p))}. \quad (59)$$

Note that $N_p(\mathbf{y})/N_{\tilde{p}}(\mathbf{y}) = \exp\left(-\frac{1}{2}\left(\mathbf{y}^T\mathbf{\Sigma}^{-1}\mathbf{y} - (\mathbf{y}^T - \boldsymbol{\mu})\mathbf{\Sigma}^{-1}(\mathbf{y} - \boldsymbol{\mu})\right)\right) =: \exp(Q(\mathbf{y}))$ and that the quotient in the integrand can be written as $q(x) = \frac{e^x c}{e^x c + (1-c)}$ for $c := P_0(p) \in (0, 1)$. Since we wish to lower bound the expectation of $P_{t+1}(p)$, we will utilize the lower bounds $q(x) \geq ce^x$ $\forall x \leq 0$ and $q(x) \geq 1 - \frac{1-c}{c}e^{-x}$ $\forall x \in \mathbb{R}$. The first lower bound will be applied to the region $H := \{\mathbf{y}\big|Q(\mathbf{y}) \leq 0\}$ whilst the second is applied to the complement $H^c$:

$$\mathbb{E}_{\mathbf{y}}\left[P_{t+1}(p)\big|x_{1:t}\right] = \frac{1}{\sqrt{(2\pi)^t}}\int_{\mathbb{R}^t}\frac{1}{\sqrt{\det\mathbf{\Sigma}}}\frac{\exp(Q(\mathbf{y}))c}{\exp(Q(\mathbf{y})c + 1 - c}\exp\left(-\frac{1}{2}\mathbf{y}^T\mathbf{\Sigma}^{-1}\mathbf{y}\right)d\mathbf{y} \quad (60)$$

$$\geq \frac{1}{\sqrt{(2\pi)^t}}\left(c\underbrace{\int_H \frac{1}{\sqrt{\det\mathbf{\Sigma}}}\exp(Q(\mathbf{y}))\exp\left(-\frac{1}{2}\mathbf{y}^T\mathbf{\Sigma}^{-1}\mathbf{y}\right)d\mathbf{y}}_{I_1 :=} \quad (61)$$

$$+ \underbrace{\int_{H^c}\frac{1}{\sqrt{\det\mathbf{\Sigma}}}\left(1 - \frac{1-c}{c}\exp(-Q(\mathbf{y}))\right)\exp\left(-\frac{1}{2}\mathbf{y}^T\mathbf{\Sigma}^{-1}\mathbf{y}\right)d\mathbf{y}}_{I_2 :=}\right). \quad (62)$$

Let $\mathbf{\Sigma}^{\frac{1}{2}}$ be the positive definite and symmetric square root of $\mathbf{\Sigma}$ s.t. $\mathbf{\Sigma} = \mathbf{\Sigma}^{\frac{1}{2}}\mathbf{\Sigma}^{\frac{1}{2}}$. Next, we apply the change of variables $\mathbf{y} = \mathbf{\Sigma}^{\frac{1}{2}}(v\bar{\boldsymbol{\mu}} + \boldsymbol{w})$ s.t. $\boldsymbol{w} \perp \bar{\boldsymbol{\mu}}$ and $\bar{\boldsymbol{\mu}} := \mathbf{\Sigma}^{-\frac{1}{2}}\boldsymbol{\mu}/\|\mathbf{\Sigma}^{-\frac{1}{2}}\boldsymbol{\mu}\|$. Note that the Jacobian of this variable change is $\sqrt{\det\mathbf{\Sigma}}$ and

$$\mathbf{y}^T\mathbf{\Sigma}^{-1}\mathbf{y} = (v\bar{\boldsymbol{\mu}} + \boldsymbol{w})^T\mathbf{\Sigma}^{\frac{1}{2}}\mathbf{\Sigma}^{-1}\mathbf{\Sigma}^{\frac{1}{2}}(v\bar{\boldsymbol{\mu}} + \boldsymbol{w}) = \|v\bar{\boldsymbol{\mu}} + \boldsymbol{w}\|^2 = v^2 + \|\boldsymbol{w}\|^2, \quad (63)$$

$$(\mathbf{y} - \boldsymbol{\mu})^T\mathbf{\Sigma}^{-1}(\mathbf{y} - \boldsymbol{\mu}) = ((v - \|\mathbf{\Sigma}^{-\frac{1}{2}}\boldsymbol{\mu}\|)\bar{\boldsymbol{\mu}} + \boldsymbol{w})^T((v - \|\mathbf{\Sigma}^{-\frac{1}{2}}\boldsymbol{\mu}\|)\bar{\boldsymbol{\mu}} + \boldsymbol{w}) \quad (64)$$

$$= (v - \|\mathbf{\Sigma}^{-\frac{1}{2}}\boldsymbol{\mu}\|)^2 + \|\boldsymbol{w}\|^2, \quad (65)$$

$$Q(\mathbf{y}) = -\frac{1}{2}\left(v^2 + \|\boldsymbol{w}\|^2 - (v - \|\mathbf{\Sigma}^{-\frac{1}{2}}\boldsymbol{\mu}\|)^2 - \|\boldsymbol{w}\|^2\right) = -\frac{1}{2}\left(2v\|\mathbf{\Sigma}^{-\frac{1}{2}}\boldsymbol{\mu}\| - \|\mathbf{\Sigma}^{-\frac{1}{2}}\boldsymbol{\mu}\|^2\right). \quad (66)$$

The region $H = \{Q(\mathbf{y}) \leq 0\}$ can then be expressed as $H = \left\{(v, \boldsymbol{w})\big|v \geq \|\mathbf{\Sigma}^{-\frac{1}{2}}\boldsymbol{\mu}\|/2, \boldsymbol{w} \perp \boldsymbol{\mu}\right\}$ and $H^c = \{(v, \boldsymbol{w})\big|v < \|\mathbf{\Sigma}^{-\frac{1}{2}}\boldsymbol{\mu}\|/2, \boldsymbol{w} \perp \boldsymbol{\mu}\}$. Next, we apply the change of variables to $I_1$ and

decompose the integral into two parts:

$$I_1 = \int_H \exp\left(-\frac{1}{2}\left(-\|\boldsymbol{\Sigma}^{-\frac{1}{2}}\boldsymbol{\mu}\|^2 + 2v\|\boldsymbol{\Sigma}^{-\frac{1}{2}}\boldsymbol{\mu}\| + v^2 + \|\boldsymbol{w}\|^2\right)\right)dvd\boldsymbol{w} \tag{67}$$

$$= \exp(\|\boldsymbol{\Sigma}^{-\frac{1}{2}}\boldsymbol{\mu}\|^2)\int_{v\geq\|\boldsymbol{\Sigma}^{-\frac{1}{2}}\boldsymbol{\mu}\|/2} \exp\left(-\frac{1}{2}(v+\|\boldsymbol{\Sigma}^{-\frac{1}{2}}\boldsymbol{\mu}\|)^2\right)dv \int_{\mathbb{R}^{t-1}} \exp\left(-\frac{\|\boldsymbol{w}\|^2}{2}\right)d\boldsymbol{w} \tag{68}$$

$$= \exp(\|\boldsymbol{\Sigma}^{-\frac{1}{2}}\boldsymbol{\mu}\|^2)\cdot\sqrt{2\pi}\left(1-\Phi\left(\frac{3}{2}\|\boldsymbol{\Sigma}^{-\frac{1}{2}}\boldsymbol{\mu}\|\right)\right)\cdot\sqrt{(2\pi)^{t-1}} \tag{69}$$

$$= \sqrt{(2\pi)^t}\exp(\|\boldsymbol{\Sigma}^{-\frac{1}{2}}\boldsymbol{\mu}\|^2)\Phi\left(-\frac{3}{2}\|\boldsymbol{\Sigma}^{-\frac{1}{2}}\boldsymbol{\mu}\|\right) \qquad (\Phi(-x)=1-\Phi(x)) \tag{70}$$

where $\Phi(\cdot)$ is the CDF of the unit-Gaussian. Similarly, we apply the change of variables to $I_2$ and decompose the integral into four parts:

$$I_2 = \int_{H^c}\left(\exp\left(-\frac{1}{2}\left(v^2+\|\boldsymbol{w}\|^2\right)\right)\right. \tag{71}$$

$$\left. -\frac{1-c}{c}\exp\left(-\frac{1}{2}(-2v\|\boldsymbol{\Sigma}^{-\frac{1}{2}}\boldsymbol{\mu}\| + \|\boldsymbol{\Sigma}^{-\frac{1}{2}}\boldsymbol{\mu}\|^2 + v^2 + \|\boldsymbol{w}\|^2)\right)\right)dvd\boldsymbol{w} \tag{72}$$

$$= \int_{v<\|\boldsymbol{\Sigma}^{-\frac{1}{2}}\boldsymbol{\mu}\|/2}\exp\left(-\frac{v^2}{2}\right)dv\int_{\mathbb{R}^{t-1}}\exp\left(-\frac{w^2}{2}\right)d\boldsymbol{w} \tag{73}$$

$$-\frac{1-c}{c}\int_{v<\|\boldsymbol{\Sigma}^{-\frac{1}{2}}\boldsymbol{\mu}\|/2}\exp\left(-\frac{1}{2}\left(v-\|\boldsymbol{\Sigma}^{-\frac{1}{2}}\boldsymbol{\mu}\|\right)^2\right)dv\int_{\mathbb{R}^{t-1}}\exp\left(-\frac{\|\boldsymbol{\Sigma}^{-\frac{1}{2}}\boldsymbol{\mu}\|^2}{2}\right)d\boldsymbol{w} \tag{74}$$

$$= \sqrt{2\pi}\Phi\left(\frac{\|\boldsymbol{\Sigma}^{-\frac{1}{2}}\boldsymbol{\mu}\|}{2}\right)\cdot\sqrt{(2\pi)^{t-1}} - \frac{1-c}{c}\sqrt{2\pi}\Phi\left(-\frac{\|\boldsymbol{\Sigma}^{-\frac{1}{2}}\boldsymbol{\mu}\|}{2}\right)\cdot\sqrt{(2\pi)^{t-1}} \tag{75}$$

$$= \sqrt{(2\pi)^t}\left(1-\frac{1}{c}\Phi\left(-\frac{\|\boldsymbol{\Sigma}^{-\frac{1}{2}}\boldsymbol{\mu}\|}{2}\right)\right). \qquad (\Phi(-x)=1-\Phi(x)) \tag{76}$$

Combining the results for $I_1$ and $I_2$, we get that

$$\mathbb{E}_{\mathbf{y}}\left[P_{t+1}(p)\big|x_{1:t}\right] \geq 1 + P_0(p)e^{\|\boldsymbol{\Sigma}^{-\frac{1}{2}}\boldsymbol{\mu}\|^2}\Phi\left(-\frac{3}{2}\|\boldsymbol{\Sigma}^{-\frac{1}{2}}\boldsymbol{\mu}\|\right) - \frac{1}{P_0(p)}\Phi\left(-\frac{\|\boldsymbol{\Sigma}^{-\frac{1}{2}}\boldsymbol{\mu}\|}{2}\right) \tag{77}$$

$$\square$$

Note that a sharp bound of Eq. (77) in terms of elementary functions can be obtained using bounds on the error function (Cook, 2018; Abramowitz & Stegun, 1972):

$$\mathbb{E}_{\mathbf{y}}\left[P_{t+1}(p)\big|x_{1:t}\right] \geq 1 + P_0(p)\sqrt{\frac{2}{\pi}}e^{-\frac{\|\boldsymbol{\Sigma}^{-\frac{1}{2}}\boldsymbol{\mu}\|^2}{8}}\frac{2}{3\|\boldsymbol{\Sigma}^{-\frac{1}{2}}\boldsymbol{\mu}\| + \sqrt{9\|\boldsymbol{\Sigma}^{-\frac{1}{2}}\boldsymbol{\mu}\|^2 + 16}} \tag{78}$$

$$-\frac{1}{P_0(p)}\sqrt{\frac{2}{\pi}}e^{-\frac{\|\boldsymbol{\Sigma}^{-\frac{1}{2}}\boldsymbol{\mu}\|^2}{8}}\frac{2}{\|\boldsymbol{\Sigma}^{-\frac{1}{2}}\boldsymbol{\mu}\| + \sqrt{\|\boldsymbol{\Sigma}^{-\frac{1}{2}}\boldsymbol{\mu}\|^2 + 32\pi^{-1}}}. \tag{79}$$

## A.4 INFORMATION-THEORETIC REGRET BOUND FOR GP-TS

We begin by showing that the rewards are subgaussian.

**Lemma A.4.** *Fix $x \in \mathcal{X}$ and let $Z = f(x) + \epsilon_t - \mathbb{E}_t[f(x)]$. If $p^* \sim P_1$, $f \sim \mathcal{GP}(\mu_{1,p^*}, k_{1,p^*})$ with $k_{1,p^*}(\cdot,\cdot) \leq \sigma_0^2$, then $Z|H_t$ is $\sqrt{\sigma_0^2 + \sigma^2}$-subgaussian.*

*Proof.* Start by considering $Z = \sum_{p \in P} \mathbb{1}\{p^* = p\}(f_p(x) - \mu_{t,p}(x))$ where $f_p \sim \mathcal{GP}_p$ such that $f_p(x) - \mu_{t,p}(x)$ is $\sigma_0$-subgaussian. Then,

$$\mathbb{E}[\lambda Z | H_t] = \mathbb{E}[\mathbb{E}[\exp(\lambda Z) \,|\, p^* = p, H_t] \tag{80}$$

$$= \sum_{p \in P} \mathbb{P}_t(p^* = p) \mathbb{E}[\exp(\lambda(f_p(x) - \mu_{t,p}(x))) \,|\, p^* = p, H_t] \tag{81}$$

$$\leq \sum_{p \in P} \mathbb{P}_t(p^* = p) \exp(\sigma_{t,p}^2(x)\lambda^2/2) \tag{82}$$

$$\leq \sum_{p \in P} \mathbb{P}_t(p^* = p) \exp(\sigma_0^2 \lambda^2/2) \tag{83}$$

$$= \exp(\sigma_0^2 \lambda/2) \tag{84}$$

Hence, $Z$ is $\sigma$-subgaussian. Then, $Z + \epsilon_t | H_t$ is $\sqrt{\sigma_0^2 + \sigma^2}$-subgaussian since $Z|H_t$ and $\epsilon_t | H_t$ are independent. □

Then, we state and prove the information-theoretic regret bound for HP-GP-TS.

**Theorem 4.6.** *If $p^* \sim P_1$, $f \sim \mathcal{GP}(\mu_{1,p^*}, k_{1,p^*})$, the Bayesian regret of HP-GP-TS is bounded by*

$$BR(T) \leq \sqrt{2|\mathcal{X}| \log(|\mathcal{X}|)(\sigma_0^2 + \sigma^2)T}. \tag{11}$$

*Proof.* Here, we analyze the regret of HP-GP-TS using the information-theoretic framework of Russo & Van Roy (2016). The proof presented here follows the proof in section D.2 of Russo & Van Roy (2016) with subgaussian noise. The idea of the information-theoretic framework is to express the regret as follows

$$\mathbb{E}\left[\sum_{t \in [T]} f(x^*) - f(x_t)\right] = \mathbb{E}\left[\sum_{t \in [T]} \mathbb{E}\big[\underbrace{f(x^*) - f(x_t)}_{\geq 0}|H_t\big]\right] \quad \text{(Tower rule)} \tag{85}$$

$$= \mathbb{E}\left[\sum_{t \in [T]} \sqrt{\mathbb{E}[f(x^*) - f(x_t)|H_t]^2 \cdot \frac{I(\cdot\,;\cdot\,|H_t)}{I(\cdot\,;\cdot\,|H_t)}}\right] \tag{86}$$

where $I(\cdot\,;\cdot\,|H_t)$ is the mutual information between two carefully chosen variables such that $\mathbb{E}[f(x^*) - f(x_t)|H_t]^2 \leq C I(\cdot\,;\cdot\,|H_t)$ for some $C$.

Let $\mathbb{E}_t[\cdot] = \mathbb{E}[\cdot|H_t], \mathbb{P}_t[\cdot] = \mathbb{P}[\cdot|H_t]$. Then,

$$\mathbb{E}_t[f(x^*) - f(x_t)] = \sum_{x \in \mathcal{X}} \mathbb{P}_t(x^* = x)\mathbb{E}_t[f(x)|x^* = x] - \sum_{x \in \mathcal{X}} \mathbb{P}_t(x_t = x)\mathbb{E}_t[f(x)|x_t = x] \tag{87}$$

$$= \sum_{x \in \mathcal{X}} \mathbb{P}_t(x^* = x)\left(\mathbb{E}_t[f(x)|x^* = x] - \mathbb{E}_t[f(x)|x_t = x]\right) \quad (\mathbb{P}_t(x_t = x) = \mathbb{P}_t(x^* = x)) \tag{88}$$

$$= \sum_{x \in \mathcal{X}} \mathbb{P}_t(x^* = x)\left(\mathbb{E}_t[f(x)|x^* = x] - \mathbb{E}_t[f(x)]\right). \quad \begin{pmatrix} \mathbb{E}_t[f(x)|x_t = x] = \mathbb{E}_t[f(x)] \\ \text{since } f|H_t \perp\!\!\!\perp x_t | H_t \end{pmatrix} \tag{89}$$

Let $Z = f(x) + \epsilon_t - \mathbb{E}_t[f(x)]$. Note that $Z|H_t$ is $\sqrt{\sigma_0^2 + \sigma^2}$-subgaussian. Consequently, $\log \mathbb{E}_t[\exp(\lambda Z)] \leq \frac{\lambda^2(\sigma_0^2 + \sigma^2)}{2} \; \forall \lambda \in \mathbb{R}$.

Fix $x^*, x \in \mathcal{X}$. Then,

$$\lambda\Big(\mathbb{E}_t[f(x) + \epsilon_t | x^* = x^*] - \mathbb{E}_t[f(x)]\Big) - \frac{\lambda^2(\sigma_0^2 + \sigma^2)}{2} \tag{90}$$

$$\leq \lambda\Big(\mathbb{E}_t[f(x) + \epsilon_t | x^* = x^*] - \mathbb{E}_t[f(x)]\Big) \tag{91}$$

$$\quad - \log \mathbb{E}_t[\exp(\lambda(f(x) + \epsilon_t - \mathbb{E}_t[f(x)]))] \qquad (Z|H_t \text{ is } \sqrt{\sigma_0^2 + \sigma^2}\text{-subgaussian}) \tag{92}$$

$$= \lambda\mathbb{E}_t[Z | x^* = x^*] - \log \mathbb{E}_t[\exp(\lambda Z)] \tag{93}$$

$$\leq D\left(\mathbb{P}_t(f(x) + \epsilon_t | x^* = x^*) || \mathbb{P}_t(f(x) + \epsilon_t)\right). \qquad \text{(Fact 12 of Russo \& Van Roy (2016))} \tag{94}$$

Now, let $\lambda = \frac{1}{\sigma_0^2 + \sigma^2} \left(\mathbb{E}_t[f(x) + \epsilon_t | x^* = x^*] - \mathbb{E}_t[f(x)]\right)$, then the following holds for all $x^*, x \in \mathcal{X}$

$$\mathbb{E}_t[f(x) | x^* = x^*] - \mathbb{E}_t[f(x)] \leq \sqrt{2(\sigma_0^2 + \sigma^2)D\left(\mathbb{P}_t(f(x) + \epsilon_t | x^* = x^*) || \mathbb{P}_t(f(x) + \epsilon_t)\right)}. \tag{95}$$

Let $I_t(\cdot;\cdot) = I(\cdot;\cdot | H_t = h_t)$. Then,

$$I_t(x^*; (x_t, y_t)) = I_t(x^*; x_t) + I_t(x^*; y_t | x_t) \qquad \text{(Chain rule)} \tag{96}$$

$$= I_t(x^*; y_t | x_t) \qquad (x^* | H_t \perp\!\!\!\perp x_t | H_t) \tag{97}$$

$$= \sum_{x \in \mathcal{X}} \mathbb{P}_t(x_t = x) I_t(x^*; y_t | x_t = x) \tag{98}$$

$$= \sum_{x \in \mathcal{X}} \mathbb{P}_t(x_t = x) I_t(x^*; f(x_t) + \epsilon_t | x_t = x) \tag{99}$$

$$= \sum_{x \in \mathcal{X}} \mathbb{P}_t(x_t = x) I_t(x^*; f(x) + \epsilon_t) \qquad (f, x^* | H_t \perp\!\!\!\perp x_t | H_t) \tag{100}$$

$$= \sum_{x \in \mathcal{X}} \mathbb{P}_t(x_t = x) \left( \sum_{x^* \in \mathcal{X}} \mathbb{P}_t(x^* = x^*) \cdot \right. \tag{101}$$

$$\left. D\left(\mathbb{P}_t(f(x) + \epsilon_t | x^* = x^*) \,\Big|\Big|\, \mathbb{P}_t(f(x) + \epsilon_t)\right) \right) \tag{102}$$

$$= \sum_{x, x^* \in \mathcal{X}} \mathbb{P}_t(x^* = x) \mathbb{P}_t(x^* = x^*) \cdot \qquad (x_t | H_t \overset{d}{=} x^* | H_t) \tag{103}$$

$$D\left(\mathbb{P}_t(f(x) + \epsilon_t | x^* = x^*) \,\Big|\Big|\, \mathbb{P}_t(f(x) + \epsilon_t)\right) \tag{104}$$

Putting the above together, we now bound $\mathbb{E}_t[f(x^*) - f(x_t)]^2$:

$$\mathbb{E}_t[f(x^*) - f(x_t)]^2 = \left( \sum_{x \in \mathcal{X}} \mathbb{P}_t(x^* = x) \left(\mathbb{E}_t[f(x) | x^* = x] - \mathbb{E}_t[f(x)]\right) \right)^2 \tag{105}$$

$$\leq |\mathcal{X}| \sum_{x \in \mathcal{X}} \mathbb{P}_t(x^* = x)^2 \left(\mathbb{E}_t[f(x) | x^* = x] - \mathbb{E}_t[f(x)]\right)^2 \qquad \text{(Cauchy-Schwarz)} \tag{106}$$

$$\leq |\mathcal{X}| \sum_{x, x^* \in \mathcal{X}} \mathbb{P}_t(x^* = x) \mathbb{P}_t(x^* = x^*) \left(\mathbb{E}_t[f(x) | x^* = x^*] - \mathbb{E}_t[f(x)]\right)^2 \tag{107}$$

$$\leq 2|\mathcal{X}|(\sigma_0^2 + \sigma^2) \sum_{x, x^* \in \mathcal{X}} \left( \mathbb{P}_t(x^* = x) \mathbb{P}_t(x^* = x^*) \cdot \right. \tag{108}$$

$$\left. D\left(\mathbb{P}_t(f(x) + \epsilon_t | x^* = x^*) || \mathbb{P}_t(f(x) + \epsilon_t)\right) \right) \tag{109}$$

$$\leq 2|\mathcal{X}|(\sigma_0^2 + \sigma^2) I_t(x^*; (x_t, y_t)). \tag{110}$$

Returning to the full regret,

$$\mathbb{E}\left[\sum_{t\in[T]} f(x^*) - f(x_t)\right] = \mathbb{E}\left[\sum_{t\in[T]}\sqrt{\mathbb{E}_t\left[f(x^*)-f(x_t)\right]^2 \cdot \frac{I_t(x^*\,;(x_t,y_t))}{I_t(x^*\,;(x_t,y_t))}}\right] \tag{111}$$

$$\leq \sqrt{2|\mathcal{X}|(\sigma_0^2+\sigma^2)}\mathbb{E}\left[\sum_{t\in[T]}\sqrt{I_t(x^*\,;(x_t,y_t))}\right] \tag{112}$$

$$\tag{113}$$

Then, the expectation can then be bounded as

$$\mathbb{E}\left[\sum_{t\in[T]}\sqrt{I_t(x^*\,;(x_t,y_t))}\right] = \leq \sum_{t\in[T]}\mathbb{E}\left[\sqrt{I_t(x^*\,;(x_t,y_t))}\right] \tag{114}$$

$$\leq \sum_{t\in[T]} 1\cdot\sqrt{\mathbb{E}\left[I_t(x^*\,;(x_t,y_t))\right]} \qquad \text{(Jensen's inequality)} \tag{115}$$

$$= \sum_{t\in[T]}\sqrt{\mathbb{E}_{H_t}\left[I(x^*\,;(x_t,y_t)|H_t = h_t)\right]} \tag{116}$$

$$= \sum_{t\in[T]}\sqrt{I(x^*\,;(x_t,y_t)|H_t)} \tag{117}$$

$$\leq \sqrt{\sum_{t\in[T]} 1^2 \sum_{t\in[T]} I(x^*\,;(x_t,y_t)|H_t)} \qquad \text{(Cauchy-Schwarz)} \tag{118}$$

Finally, the regret can be bounded as

$$\mathbb{E}\left[\sum_{t\in[T]} f(x^*) - f(x_t)\right] = \sqrt{2|\mathcal{X}|(\sigma_0^2+\sigma^2)}\sqrt{TI(x^*;H_T)} \tag{119}$$

$$\leq \sqrt{2|\mathcal{X}|(\sigma_0^2+\sigma^2)}\sqrt{TH(x^*)} \tag{120}$$

$$\leq \sqrt{2|\mathcal{X}|(\sigma_0^2+\sigma^2)}\sqrt{T\log|\mathcal{X}|} \tag{121}$$

$$\square$$

## B    DESCRIPTION OF KERNELS

The RBF kernel, $k(x,\tilde{x}) = \exp(-||x-\tilde{x}||^2/\ell^2)$ guarantees that $f$ is smooth. The length-scale parameter $\ell > 0$ determines how quickly $f$ changes, smaller values lead to more fluctuations. The rational quadratic (RQ) kernel $k(x,\tilde{x}) = \left(1 + \frac{||x-\tilde{x}||^2}{2\alpha\ell^2}\right)^{-\alpha}$ where $\alpha > 0$ is a mixture of RBF kernels with varying lengthscales. The Matérn kernel (Matérn, 1986) $k(x,\tilde{x}) = \frac{2^{1-\nu}}{\Gamma(\nu)}\left(\frac{\sqrt{2\nu}||x-\tilde{x}||}{\ell}\right)^\nu K_\nu\left(\frac{\sqrt{2\nu}||x-\tilde{x}||}{\ell}\right)$ where $\nu > 0$ is the smoothness parameter that imposes that $f$ is k-times differentiable if $\nu > k$ for integer $k$. The functions $\Gamma(\nu)$ and $K_v$ correspond to the gamma function and a modified Bessel function (Williams & Rasmussen, 2006). The periodic kernel $k(x,\tilde{x}) = \exp\left(-\frac{1}{2}\sum_{i=1}^d \sin^2(\frac{\pi}{\rho}(x-\tilde{x}))/\ell\right)$ generates smooth and periodic functions with period $\rho > 0$ (Mackay, 1998). The linear kernel $k(x,\tilde{x}) = vx^\top\tilde{x}/\nu$ generates linear functions where $v$ is the variance parameter.

## C    ADDITIONAL EXPERIMENTAL DETAILS

In this section, we provide some additional details about the experiments.

For all the real-world datasets, sensors containing any null measurements have been filtered out.

The Intel Berkeley dataset consists of measurements from 46 temperature sensors across 19 days. The training set consists of the first 12 days of measurements and the remaining 7 days constitute the test set. The noise variance is set to $\sigma^2 = 0.7^2$.

The PeMS data consists of measurements from 211 sensors along the I-880 highway from all of 2023. The goal is to find the sensors with low speeds to identify congestions. We use the 5-min averages provided by PeMS. Data between 2023-01-01 and 2023-09-01 is put into the training set whilst the data until 2023-12-31 is put into the test set. The noise variance is set to $\sigma^2 = 2.25^2$.

The PNW precipication data consists of daily precipitation data from 1949 to 1950 across $167\ 50 \times 50$ km regions in the Pacific Northwest. The goal is to find the region with the highest precipitation for any given day. The training data consists of the measurements made prior to 1980 and the test data consists of the measurements between 1980 and 1994. The original data is stated to be given in mm/day however the data seems to be off by a factor of 10. We rescale the data to a log-scale using $\log(\cdot/10 + 0.1)$, similar to Krause et al. (2008). The noise variance is set to $\sigma^2 = 0.41^2$.

In the Intel experiment, we removed one outlier seed. All methods had a final cumulative regret around 6000°C, note that the average for the worst performing model across the other seeds was $\approx 150$°C. The outlier is shown Fig. 7. We can see that one of the sensors display very high temperatures compared to all other sensors, which is why all methods performed poorly on this seed. It should be noted that many of the sensors in the Intel data logged degrees above 100°C after a certain time - likely due to sensor failure rather boiling temperatures in an office environment. Also note that these days were excluded from both our training and test data. The outlier could be an indication that this particular sensor was starting to fail earlier than others.

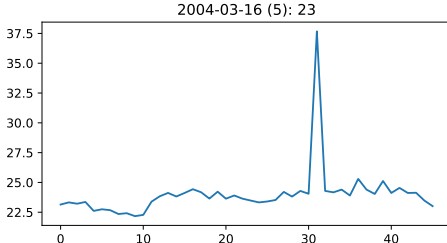

Figure 7: Removed sample from the test data in the Intel experiment. One of the sensors display very high temperatures.

## D  ADDITIONAL EXPERIMENTAL RESULTS

In this section, we provide some additional experimental results.

First, we include the mean number of priors in $P_t$ for all experiments in Fig. 8. Similarly, we include the average entropy of the hyperposterior for all experiments in Fig. 9. For the lengthscale, subspace, PeMS and PNW precipication experiments, hardly any priors are eliminated. In contrast, the hyperposterior entropy concentrates rapidly across all experiments with the subspace and PNW precipication having the most and least concentrated hyperposterior.

The cost of learning the prior, $\sum_{t \in [T]} \mathbb{E}[U_{t,p^*}(x^*) - U_{t,p_t}(x^*)]$, for HP-GP-TS is shown in Fig. 10. Whilst this term has not been bounded theoretically in Theorem 4.5, we observe that empirically its growth diminishes much quicker than the cumulative regret. Since $\sum_{t \in [T]} \mathbb{E}[U_{t,p^*}(x^*) - U_{t,p_t}(x^*)]$ is one term in the upper bound for the regret of HP-GP-TS, learning the prior does not seem to increase the growth rate of the regret empirically.

We include the full set of confusion matrices for the lengthscale and subspace experiments in Fig. 11. In the lengthscale experiments, we observe that PE-GP-UCB and -TS oversample the shortest lengthscale. This is similar to the kernel experiment where the Matérn 3/2 kernel was also oversampled. However, we see that HP-GP-TS and MAP GP-TS do not suffer from this optimistic bias. In the subspace experiment, HP- and MAP GP-TS have an accuracy of around 96% where as

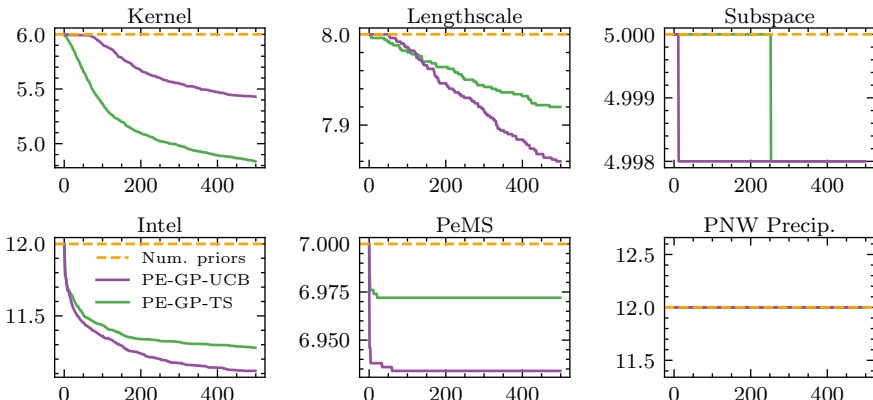

Figure 8: Mean number of priors remaining in $P_t$ over time for PE-GP-UCB and -TS.

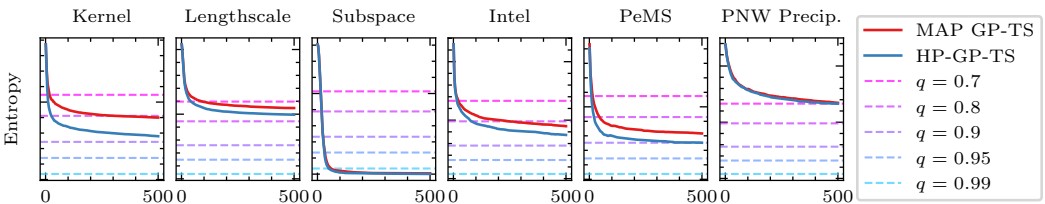

Figure 9: Average entropy in the hyperposterior $P_t$ over time for HP- and MAP GP-TS. The dashed reference values correspond to entropies of discrete distributions with prob. $q$ on one choice and prob. $\frac{1-q}{|P|-1}$ on the other $|P|-1$ choices.

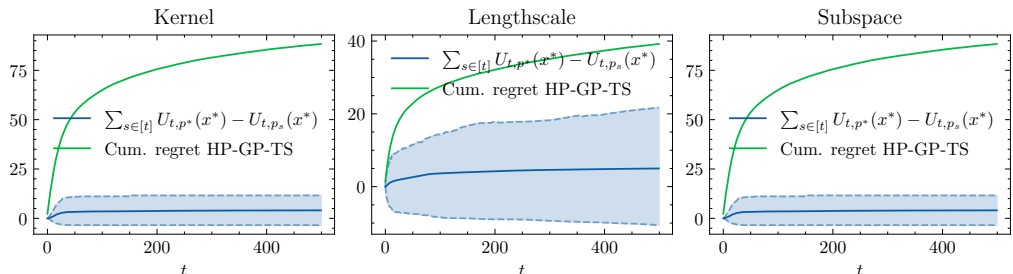

Figure 10: The cost of learning the prior, $\sum_{t \in [T]} U_{t,p^*}(x^*) - U_{t,p_t}(x^*)$, for HP-GP-TS in the synthetic experiments. The solid blue line corresponds to the mean and the dashed lines corresponds to the first and last decile at each time step.

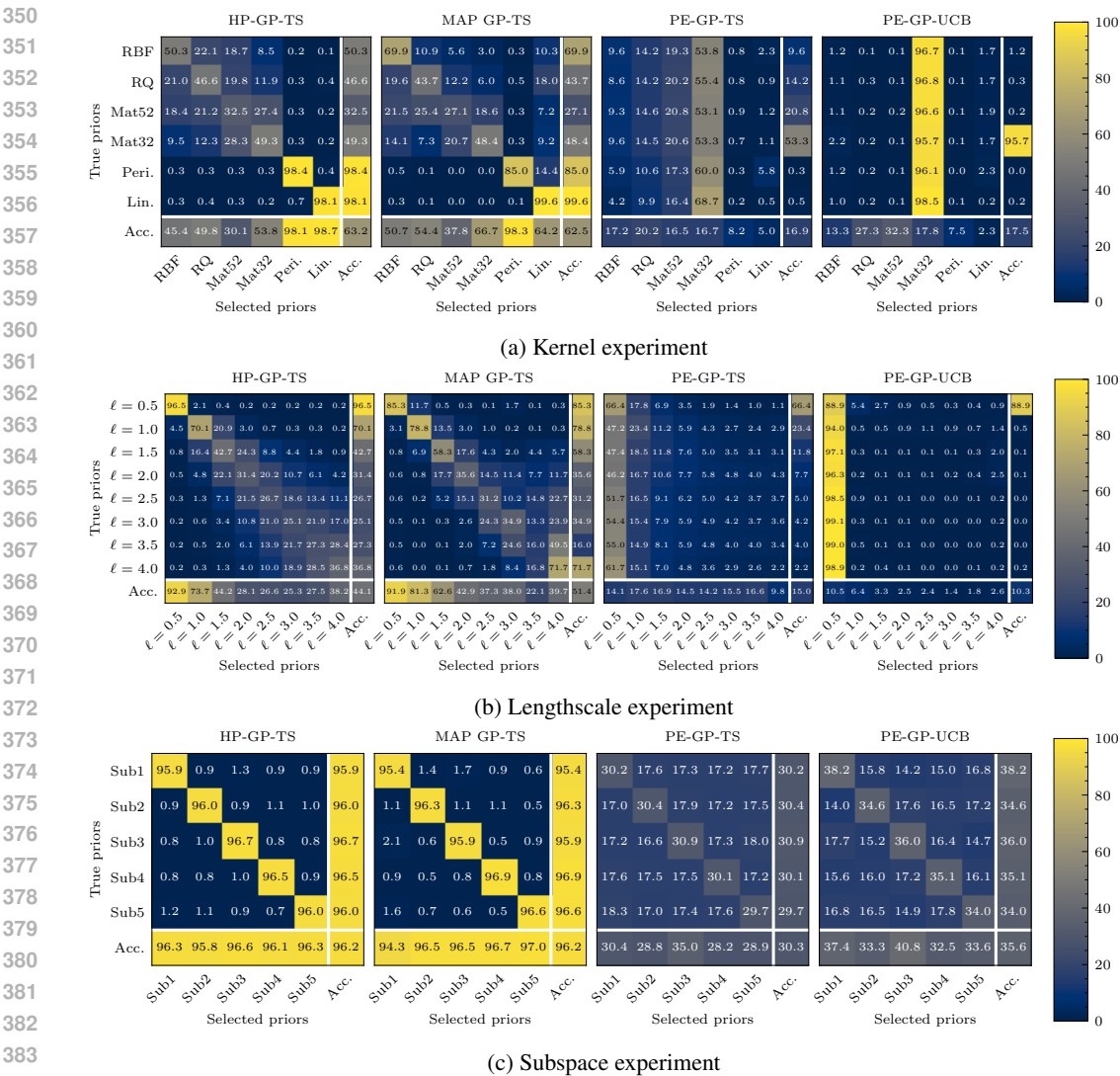

(a) Kernel experiment

(b) Lengthscale experiment

(c) Subspace experiment

Figure 11: Confusion matrices for the true prior $p^*$ and $p_t$ across all time steps of the synthetic experiments.

PE-GP-TS and -UCB have accuracies 30% and 36% respectively. The priors are equivalent up to coordinate permutations and are therefore difficult. The PE-methods do not oversample any specific prior but commit to much time to exploring along the irrelevant dimensions.

In Tables 1 and 2, the total regret for the lengthscale and subspace scaling experiments are shown.

In Fig. 12, we visualize the median cumulative regret on the real-world data experiments. Similarly, in Fig. 13, we show further quantiles of the final cumulative regret. MAP and HP-GP-TS exhibit similar median regret across the experiments but MAP GP-TS has a larger variance and tail values.

Table 1: Average total regret and $\pm 1$ standard error for the lengthscale experiment as $|P|$ increases.

| Algorithm | Lengthscales, $|P|$ | | | | |
|---|---|---|---|---|---|
| | 8 | 16 | 32 | 64 | 128 |
| MAP GP-TS | $30.2 \pm 1.2$ | $32.4 \pm 2.5$ | $32.5 \pm 2.1$ | $28.7 \pm 1.1$ | $30.8 \pm 1.9$ |
| HP-GP-TS | $31.4 \pm 1.0$ | $31.7 \pm 0.9$ | $30.8 \pm 0.8$ | $30.7 \pm 1.0$ | $31.0 \pm 1.4$ |
| PE-GP-TS | $61.8 \pm 0.5$ | $61.3 \pm 0.5$ | $62.2 \pm 0.5$ | $62.4 \pm 0.4$ | $64.3 \pm 0.4$ |
| PE-GP-UCB | $114.2 \pm 0.6$ | $114.8 \pm 0.6$ | $115.5 \pm 0.6$ | $114.5 \pm 0.6$ | $114.8 \pm 0.6$ |
| Oracle GP-TS | $28.1 \pm 0.8$ | $26.4 \pm 0.8$ | $27.3 \pm 0.8$ | $26.5 \pm 0.7$ | $25.7 \pm 0.7$ |
| Oracle GP-UCB | $48.3 \pm 1.2$ | $46.9 \pm 1.1$ | $48.4 \pm 1.1$ | $46.5 \pm 1.0$ | $45.6 \pm 1.0$ |

Table 2: Average total regret and $\pm 1$ standard error for the subspace experiment as $|P|$ increases.

| Algorithm | Subspaces, $|P|$ | | | |
|---|---|---|---|---|
| | 5 | 8 | 12 | 16 |
| MAP GP-TS | $87.2 \pm 1.0$ | $89.9 \pm 1.1$ | $89.1 \pm 0.9$ | $90.9 \pm 1.2$ |
| HP-GP-TS | $88.3 \pm 0.9$ | $88.8 \pm 0.9$ | $89.5 \pm 0.9$ | $90.8 \pm 0.9$ |
| PE-GP-TS | $177.1 \pm 1.4$ | $269.5 \pm 1.9$ | $344.7 \pm 2.3$ | $396.9 \pm 2.5$ |
| PE-GP-UCB | $389.0 \pm 1.5$ | $526.0 \pm 1.8$ | $622.4 \pm 2.3$ | $688.0 \pm 2.7$ |
| Oracle GP-TS | $86.0 \pm 1.0$ | $84.1 \pm 0.9$ | $84.6 \pm 1.0$ | $84.8 \pm 1.0$ |
| Oracle GP-UCB | $217.3 \pm 1.0$ | $218.2 \pm 1.0$ | $218.6 \pm 1.0$ | $218.9 \pm 0.9$ |

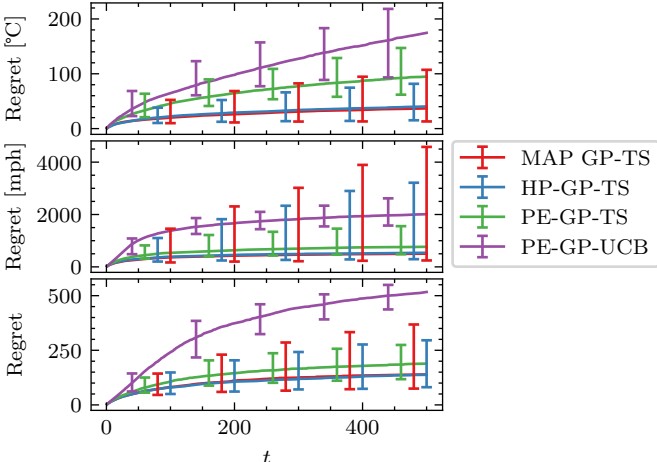

Figure 12: Median cumulative regret on Intel temperature data (top), PeMS speed data (middle) and PNW precipication data (bottom). Errorbars correspond to first and last decile.

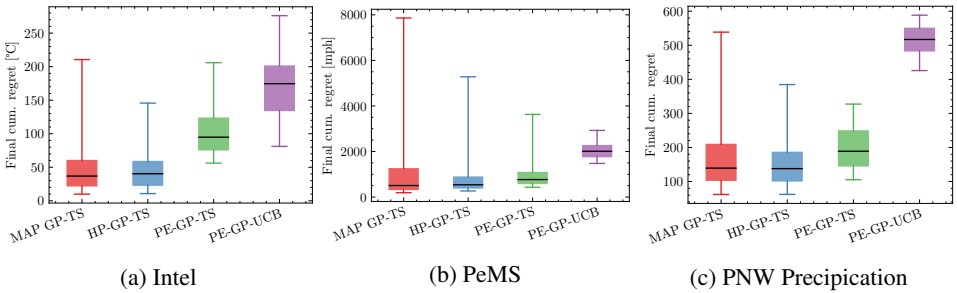

(a) Intel      (b) PeMS      (c) PNW Precipitation

Figure 13: Quantiles of the final cumulative regret on the real-world data experiments. The median is shown with a black line. The whiskers correspond to the 5th and 95th percentile and the lower and upper edges of the boxes show the first and third quantile.

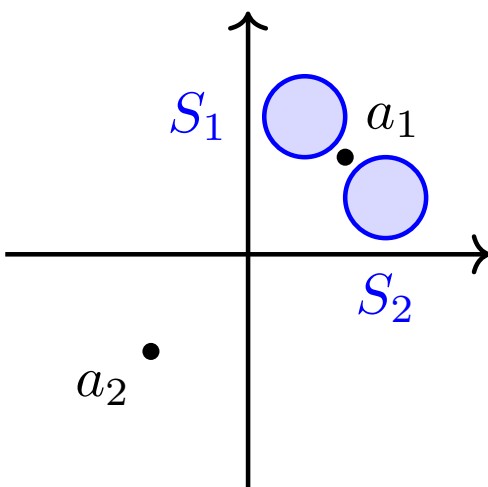

Figure 14: Potential counterexample to Eq (9) in Hong et al. (2022b). The blue regions represent when the event $E_0$ holds and the black dots represent the two arms $a_1$ and $a_2$.

# E    TECHNICAL ISSUES IN HONG ET AL. (2022B)

Theorem 1 in Hong et al. (2022b) provides a regret bound for MixTS in the linear setting. The linear setting assumes that the true parameter $\theta^* \sim \mathcal{N}(\theta_{0,S_*}, \Sigma_{0,S_*})$ where the latent state $S_*$ is sampled from a discrete mixture prior $P_0$. The proof of Theorem 1 in Hong et al. (2022b) contains a few non-obvious steps that seem to difficult to motivate. We use the notation of Hong et al. (2022b).

1. Eq. (9) in Hong et al. (2022b) uses the TS property that the true prior and optimal arm is equal in distribution to the selected prior and selected arm given the history: $A_{t,*}^\top \bar{\theta}_{t,S_*} | H_t \stackrel{d}{=} A_t^\top \bar{\theta}_{t,S_t} | H_t$ (equivalent to $\mu_{t,p^*}(x^*) | H_t \stackrel{d}{=} \mu_{t,p_t}(x_t) | H_t$ in our notation). However, Eq. (9) additionally conditions on the event $E_0 = \{\|\theta_* - \theta_{0,S_*}\|_{\Sigma_{0,S_*}^{-1}} \le \sqrt{2d \log(dn)}\}$ where $\theta_*$ lies close to its prior mean. It is unclear if the TS property holds under this event. Consider the example in Fig. 14, if $E_0$ holds then $\theta_*$ lies in the blue regions and thus $a_2$ is optimal w.p. 0. If $E_0^c$ holds, then $a_2$ is optimal with a non-zero probability. However, MixTS is oblivious to $E_0$ given the history and thus $A_{t,*} | H_t, E_0 \stackrel{d}{\ne} A_t | H_t$. This counterexample illustrates the overall idea but we have not validated that the scale of the arms and the blue regions are feasible.

2. Five lines above Eq. (9) in Hong et al. (2022b), it is stated that the *regret* is upper-bounded by a constant $M$ whenever $E_0$ occurs. However, from Eq (9) to the first term in step 3 of their analysis (page 15), the bound of $M$ is applied implicitly to $A_t^\top \bar{\theta}_{t,S_t} - A_t^\top \theta_*$ without motivation.

3. Additionally, the second term in Eq. (9) contains the indicator function $\mathbf{1}\{E_0\}$: $\mathbb{E}[(A_t^\top \bar{\theta}_{t,S_t} - A_t^\top \theta_*)\mathbf{1}\{E_0\}]$. On page 15, this indicator function is dropped without motivation: $\mathbb{E}[\langle A_t^\top \bar{\theta}_{t,S_t} - A_t^\top \theta_* \rangle_M]$ where $\langle \cdot \rangle_M = \min(\cdot, M)$ for the bound $M$. If the expression inside is non-negative w.p. 1, then this could be seen as an upper bound but it is unclear if the remaining steps would follow.

4. From our understanding, the final equation on page 15 adds and subtracts the confidence bound and adds a zero-mean Gaussian inside a minimum. However, adding a zero-mean Gaussian inside a minimum reduces the expectation but the analysis seem to assume that it would increase the expectation. I.e. it is seemingly assumed that $\mathbb{E}[\min(M, X)] \le \mathbb{E}[\min(M, X + \epsilon_t)]$ for a constant M and random variable $X$. However, the reverse inequality is true.

## F EXTENDING THE ANALYSIS OF HONG ET AL. (2022B) TO HP-GP-TS

In this section, we explore how the analysis of Hong et al. (2022b) can be applied to the Gaussian process setting considered in this paper. The core idea of the analysis of Hong et al. (2022b) is to introduce confidence intervals for the priors (or latent variables in their terminology). The confidence intervals closely resemble the prior elimination criteria introduced by Ziomek et al. (2025). In our notation, the equivalent confidence set $C_t$ of Hong et al. (2022b) is defined as

$$C_t = \left\{ p \in P : \sum_{s=1}^{t-1} \mathbb{1}\{p_s = p\}(\mu_{s,p}(x_s) - \sqrt{\beta_T}\sigma_{s,p_s}(x_s) - y_s) \leq 2\sigma\sqrt{N_t(p)\log T} \right\}. \quad (122)$$

We define $G_t(p) = \sum_{s=1}^{t-1} \mathbb{1}\{p_s = p\}(\mu_{s,p}(x_s) - \sqrt{\beta_T}\sigma_{s,p_s}(x_s) - y_s)$. The prior elimination criteria for PE-GP-UCB and PE-GP-TS can be expressed as a confidence set in the following way:

$$\tilde{C}_t = \left\{ p \in P : \forall s < t \quad \left| \sum_{\substack{i \leq s \\ p_i = p}} y_i - \mu_{i,p}(x_i) \right| \leq \sqrt{2\sigma^2 N_t(p)\log(t^2|P|\pi^2/3)} + \sum_{\substack{i \leq s \\ p_i = p}} \sqrt{\beta_i}\sigma_{i,p}(x_i) \right\}. \quad (123)$$

The main difference between $C_t$ and $\tilde{C}_t$ is that $\tilde{C}_t$ is monotonically decreasing in cardinality since eliminated priors are cannot be selected. Additionally, $\tilde{C}_t$ uses a symmetric criteria and the confidence parameters in $C_t$ use the final time step $T$ rather than the current time step $t$.

To apply the analysis of Hong et al. (2022b) directly to a GP-setting would require resolving the issues discussed in Section E. To circumvent some of these issues, we attempt to analyze the regret of HP-GP-TS by directly applying a UCB-decomposition and manipulate the decomposition such that $G_t(p_t)$ appears.

Recall, the regular UCB-decomposition to the Bayesian cumulative regret:

$$\text{BR}(T) = \sum_{t \in [T]} \mathbb{E}[f(x^*) - f(x_t)] \quad (124)$$

$$= \sum_{t \in [T]} \mathbb{E}[f(x^*) - U_{t,p^*}(x^*) + U_{t,p^*}(x^*) - f(x_t)] \quad (125)$$

$$= \underbrace{\sum_{t \in [T]} \mathbb{E}[f(x^*) - U_{t,p^*}(x^*)]}_{(1)} + \underbrace{\sum_{t \in [T]} \mathbb{E}[U_{t,p_t}(x_t) - f(x_t)]}_{(2)} \quad (p_t, x_t|H_t \stackrel{d}{=} p^*, x^*|H_t)$$

$$(126)$$

By the same steps as in the proof of Theorem 4.5, $(1) \leq \pi^2/6$. Hence, we focus on the second term and condition on whether the selected prior lies in the confidence set:

$$(2) = \sum_{t \in [T]} \mathbb{E}[(U_{t,p_t}(x_t) - f(x_t))\mathbb{1}\{p_t \in C_t\}] + \sum_{t \in [T]} \mathbb{E}[(U_{t,p_t}(x_t) - f(x_t))\mathbb{1}\{p_t \notin C_t\}]. \quad (127)$$

To bound the right-term above, we would like $(U_{t,p_t}(x_t) - f(x_t))$ to be bounded such that it is sufficient to analyze the probability $\mathbb{1}\{p_t \notin C_t\}$. However, we have not identified a method of doing so. We continue to study the left term. Since the noise $\epsilon_t$ is independent of $\mathbb{1}\{p_t \in C_t\}$ and has zero

mean, we can add it inside the expectation and add-and-subtract the confidence radius:

$$\sum_{t\in[T]} \mathbb{E}[(U_{t,p_t}(x_t) - f(x_t))\mathbb{1}\{p_t \in C_t\}] \tag{128}$$

$$= \sum_{t\in[T]} \mathbb{E}[(U_{t,p_t}(x_t) - f(x_t) - \epsilon_t)\mathbb{1}\{p_t \in C_t\}] \tag{129}$$

$$= \sum_{t\in[T]} \mathbb{E}[(\mu_{t,p_t}(x_t) - \sqrt{\beta_t}\sigma_{t,p_t}(x_t) - f(x_t) - \epsilon_t)\mathbb{1}\{p_t \in C_t\}] \tag{130}$$

$$+ \sum_{t\in[T]} 2\mathbb{E}[\sqrt{\beta_t}\sigma_{t,p_t}(x_t)\mathbb{1}\{p_t \in C_t\}] \tag{131}$$

$$\leq \underbrace{\sum_{t\in[T]} \mathbb{E}[(\mu_{t,p_t}(x_t) - \sqrt{\beta_t}\sigma_{t,p_t}(x_t) - f(x_t) - \epsilon_t)\mathbb{1}\{p_t \in C_t\}]}_{(3)} \tag{132}$$

$$+ \sum_{t\in[T]} 2\mathbb{E}[\sqrt{\beta_t}\sigma_{t,p_t}(x_t)] \qquad (\sqrt{\beta_t}\sigma_{t,p_t}(x_t) \geq 0) \tag{133}$$

Note that the sum of $\sqrt{\beta_t}\sigma_{t,p_t}(x_t)$ can be bounded in terms of the worst-case MIG $\hat{\gamma}_T$ Lemma A.3. To bound (3), following the argument of Hong et al. (2022b), we note that for each prior $p \in P$ there must be some last time $t'(p) \leq T$ that $p$ was in the confidence set and chosen by the algorithm: $p \in C_{t'}$ and $p = p_{t'}$. Since $p \in C_{t'}$, then $G_{t'}(p) \leq 2\sigma\sqrt{N_{t'}(p)\log T}$. Therefore, for each prior $p$, we can split the sum into $G_{t'}(p)$ and the final term at time $t'$:

$$(3) \leq \mathbb{E}\left[\sum_{p\in P}(G_{t'}(p) + (\mu_{t',p}(x_{t'}) - \sqrt{\beta_{t'}}\sigma_{t',p}(x_{t'}) - f(x_{t'}) - \epsilon_{t'}))\mathbb{1}\{p \in C_{t'}\}\right] \tag{134}$$

$$\leq \mathbb{E}\left[\sum_{p\in P} 2\sigma\sqrt{N_{t'}(p)\log T}\right] \tag{135}$$

$$+ \mathbb{E}\left[\sum_{p\in P}(\mu_{t',p}(x_{t'}) - \sqrt{\beta_{t'}}\sigma_{t',p}(x_{t'}) - f(x_{t'}) - \epsilon_{t'}))\mathbb{1}\{p \in C_{t'}\}\right]. \tag{136}$$

The first sum above can be bounded using that $\sum_{p\in P} N_T(p) = T$ and Jensen's inequality. The major challenge is to show that the final term is bounded. This requires creating an event s.t. the terms inside are bounded with high probability. However, as the issues in Section E highlight, if the event is in introduced at an earlier step then it must not invalidate the subsequent manipulations.

