# OpenReview forum: "Efficient Prior Selection in Gaussian Process Bandits with Thompson Sampling"
_ICLR.cc/2026/Conference — Submitted to ICLR 2026_

### Official Review · Reviewer_RXxU · 2025-10-29

**Soundness:** 2
**Presentation:** 3
**Contribution:** 1
**Rating:** 2
**Confidence:** 4

**Summary:**

The paper analysis two algorithms for Bayesian Optimisation with unknown prior: PE-GP-TS - a combination of Thompson Sampling and existing PE-GP-UCB and HP-GP-TS - a combination of Fully-Bayesian treatment of unknown prior and Thompson Sampling. Authors provide an incomplete regret bound for PE-GP-TS, leaving one term unbounded. They then proceeded to develop an incomplete bound on HP-GP-TS following the analysis of Srinivas et al, with one term unbounded, and another bound independent of MIG of the kernel following the analysis of van Roy and Russo. They then proceed to empirically evaluate the proposed algorithm on a number of benchmarks.

**Strengths:**

The paper address the topic of BO with unknown prior, which is an important, understudied topic.

**Weaknesses:**

**All algorithms are straightforward combinations of existing ones**

To me it seems, like PE-GP-TS is just PE-GP-UCB [3] with the acquisition function changed from UCB to Thompson Sampling, as such the novelty of that approach is marginal, at best. When it comes to HP-GP-TS, this is again, a combination of Thompson Sampling and well-known Fully Bayesian BO [2], where a hyper prior is used. In fact, Thompson Sampling with hyper prior was already proposed by prior work [1]. As such, I think the paper offers little to no novelty when it comes to proposed algorithms.

**Theoretical results not particularly strong and incomplete**

Theorem 4.3 leaves one term unbounded being the sum of predictive variances at the optimal point under the true prior. As the optimal point might never get queried, it is likely impossible to bound this term, meaning it could, in principle be linear. Although the term is inside square root, there is one more $T$ inside it, so together that would mean there is a linear $\mathcal{O}(T)$ outside of the square root, rendering the whole bound trivial.

Theorem 4.5 again leaves one crucial term unbounded, and there are further results bounding its growth. As such, this term could in principle, grow linearly and thus make the whole bound trivial. Without further analysis, I find little to no value in such a result.

Theorem 4.6 does not depend on the MIG of the kernel nor on any other properties of the kernel except for scale. This means the scaling of this bound is same no matter if the true objective was sampled from the simplest slowly-varying kernel or from an extremely complex kernel with short length-scale. As such, such a bound must be very worst-case, which is reflected in scaling with $|\mathcal{X}|$. The whole problem with choosing the right prior/kernel for BO is that if we choose prior/kernel that is too simple, then we risk functional mismatch, whereas if we choose one that is too complex, we risk slow convergence. Thus, omitting the kernel complexity from the analysis misses the point entirely .

As such, unfortunately, I believe none of the Theorems provides a useful, complete bound for any of the algorithms.

**Empirical results not particularly useful**

All the experimental settings are crafted so that there is only a finite number of priors to choose from, whereas in reality e.g. in case of unknown lengthscale, there space of possible priors corresponding to possible lengthscale is continuous and infinite. I understand that previous work [3] also considered discrete number of priors, however, it seems to me that was a "first line of work" on this topic and focused primarily on novel theoretical results. As such, I would expect the following work in the stream to "raise the bar higher", either by providing stronger theoretical guarantees (which as I discussed before I do not think the paper does) or by making the setting more realistic and easily applicable in practice, e.g. by lifting some assumptions used for theoretical analysis and comparing with the strongest empirical baselines in literature, such as MetaBO[4], transferBO[5] and self-correcting BO[6]. As I believe the paper does neither of these two things, I believe it is incremental at best.

**References**

[1] Hong, Joey, et al. "Thompson sampling with a mixture prior." International Conference on Artificial Intelligence and Statistics. PMLR, 2022.

[2] De Ath, George, Richard M. Everson, and Jonathan E. Fieldsend. "How Bayesian should Bayesian optimisation be?." Proceedings of the Genetic and Evolutionary Computation Conference Companion. 2021.

[3] Ziomek, Juliusz, Masaki Adachi, and Michael A. Osborne. "Time-Varying Gaussian Process Bandits with Unknown Prior." arXiv preprint arXiv:2402.01632 (2024).

[4] Maraval, Alexandre, et al. "End-to-end meta-bayesian optimisation with transformer neural processes." Advances in Neural Information Processing Systems 36 (2023): 11246-11260.

[5] Tighineanu, Petru, et al. "Transfer learning with gaussian processes for bayesian optimization." International conference on artificial intelligence and statistics. PMLR, 2022.

[6] Hvarfner, Carl, et al. "Self-correcting bayesian optimization through bayesian active learning." Advances in Neural Information Processing Systems 36 (2023): 79173-79199.

**Questions:**

- Why do you study standard (frequentist) regret for PE-GP-TS, but Bayesian regret for HP-GP-TS?

---

> ### Author Response · Authors · 2025-11-27
> **Response to reviewer RXxU**
>
> We thank **reviewer RXxU** for their detailed review and feedback. We agree with the reviewer that the topic of GP bandits with an unknown prior is an understudied topic. We address the comments of the reviewer below.
>
> > W1 All algorithms are straightforward combinations of existing ones
>
> Both the introduction and section 3.1 emphasize that PE-GP-TS is an extension of PE-GP-UCB that uses TS instead of UCB.
>
> Fully Bayesian BO has been proposed earlier than 2021 by Osborne et al. (2009), Benassi et al. (2011) and Snoek et al. (2012). They all approximate the expected value of an acquisition function $\alpha_p(x)$ w.r.t. the hyperposterior, $x_t=\arg\max_x \mathbb{E}_{p\sim P_t}[\alpha_p(x)]$, by using MCMC to generate multiple samples from the hyperposterior $P_t$. We go against the grain within this literature and do not compute the expected value but simply sample one value from the hyperposterior.
>
> > In fact, Thompson Sampling with hyper prior was already proposed by prior work [1].
>
> Thank you for bringing this work (Hong et al 2022b) to our attention! We were only familiar with the very similar work {\it Hierarchical Bayesian Bandits} (Hong et al. 2022a). Hong et al. (2022a) imposes a hyperprior on the mean function -- which would correspond to Bayesian model averaging (section 6.1 in Tighineaunu et al. 2022). However, Hong et al (2022b) imposes a mixture prior similar to us. We have updated 'previous work' in section 2 to include the work of Hong et al (2022b) and its relation to our work. Additionally, we have added a discussion of some non-obvious steps and issues in their proof for the linear setting to appendix E and F, see also the joint response for more details.
>
> > Theorem 4.3 leaves one term unbounded being the sum of predictive variances at the optimal point under the true prior... Theorem 4.5 again leaves one crucial term unbounded...
>
> We have updated section 4.1 and 4.2 to make it more clear that these terms have not been bounded. Additionally, we have added new results demonstrating the empirical behavior of the unbounded term for HP-GP-TS to appendix D. We observe that this unbounded term is empirically smaller and grows slower than the cumulative regret.
>
> > Empirical results not particularly useful ... I would expect the following work in the stream to "raise the bar higher" ...
>
> Our empirical evaluation improves upon the previous work by:
> 1) Analyzing why prior-elimination methods incur higher regret compared to hyperprior methods (Fig 3 and 4).
> 2) Adding experiments scaling in $|P|$. Notably, in the subspace experiment, we find that the regret of prior-elimination methods increase with $|P|$ whereas the hyperprior methods have constant regret.
> 3) Comparing against oracles that know the true prior, thus establishing lower bounds for the performance of TS and UCB methods.
>
> > All the experimental setting are crafted so that there is only a finite number of priors to choose from ...
>
> As the reviewer has highlighted, the lengthscale parameter is inherently a continuous parameter but we have discretized the lengthscale values in our experiments. The previous work on fully Bayesian BO have considered continuous lengthscale priors and have implemented their algorithms using MCMC -- this could similarly be done for HP-GP-TS. However, since our scaling experiments show that the regret for all algorithms is unaffected by the level of discretization, we do not expect there to be a meaningful difference in results if we had used continuous priors.
>
> > ... comparing with the strongest baselines in literature, such as MetaBO[4], transferBO[5] and self-correcting BO[6].
>
> TransferBO focuses on a setting where data from a related task is available and MetaBO assumes the ability to pre-train a transformer model through an RL pipeline. Our setting does not assume access to extra data nor extensive pre-training, thus those contributions are not directly applicable to our setting.
>
> Aside from PE-GP-UCB, we compare against two relevant baselines: oracle GP-TS and MAP GP-TS. The oracle knows the true prior and should reasonably provide a lower bound for the performance of any algorithm based on GP-TS. MAP GP-TS performs discrete maximum likelihood estimation which closely resembles what practitioners commonly use.
>
> Self-correcting BO uses an acquisition function that combines Joint Entropy Search (JES) and minimizing hyperparameter uncertainty. Based on Takeno et al (2025), TS outperforms JES in terms of cumulative regret and similarly in terms of simple regret for synthetic functions sampled from a known prior. Moreover, the repository linked in the paper for self-correcting BO seems to be unavailable.
>
> Shion Takeno et al. (2025). Regret Analysis of Posterior Sampling-Based Expected Improvement for Bayesian Optimization. Transactions on Machine Learning Research. https://openreview.net/forum?id=v0s9knY99c

---

> > ### Author Response · Authors · 2025-11-27
> > **Continuation of response to reviewer RXxU**
> >
> > > Why do you study standard (frequentist) regret for PE-GP-TS, but Bayesian regret for HP-GP-TS?
> >
> > PE-GP-TS does not make a Bayesian assumption about the prior which makes frequentist regret more suitable whereas the opposite holds for HP-GP-TS.

---

### Official Review · Reviewer_zenE · 2025-10-31

**Soundness:** 3
**Presentation:** 3
**Contribution:** 3
**Rating:** 8
**Confidence:** 3

**Summary:**

This paper addresses Gaussian Process (GP) bandits where the prior is unknown. To mitigate the over-exploration of existing UCB methods (e.g., PE-GP-UCB), the authors propose two Thompson Sampling (TS) algorithms. PE-GP-TS adapts prior elimination to TS. HP-GP-TS is a Bayesian approach using bi-level sampling: sampling a prior from a hyperposterior, then sampling a function.

**Strengths:**

1. HP-GP-TS outperforms PE-GP-UCB. Crucially (Fig. 5), its regret does not scale with $|P|$, a significant practical advantage.
2. The analysis shows theoretical advantages: dependence on average MIG (Thm 4.5) and a tighter $\mathcal{O}(\sqrt{T})$ bound via information-theoretic analysis (Thm 4.6).
3. Fig. 4 clearly demonstrates that UCB methods fixate on flexible priors to maximize optimism, leading to over-exploration, while TS methods identify the true prior more accurately.

**Weaknesses:**

1. for PE-GP-TS (Critical). Theorem 4.3 includes the term $\sqrt{... \sum_{t}\sigma_{t,p^{*}}^{2}(x^{*})}$. This depends on the cumulative posterior variance at the optimum $x^*$. If $x^*$ is rarely sampled, this variance remains high, implying linear regret $\mathcal{O}(T)$.
2. HP-GP-TS has a cumulative complexity of $\mathcal{O}(T^{4}|P|)$, which is prohibitive for long horizons.
3. The UCB-based bound (Thm 4.5) includes an unbounded term representing the cost of learning the prior in the general case.

**Questions:**

1. How is $\sum_{t}\sigma_{t,p^{*}}^{2}(x^{*})$ bounded in Thm 4.3? Please clarify how this avoids linear regret for PE-GP-TS.
2. Given the $\mathcal{O}(T^4|P|)$ complexity, are there approximations to mitigate the computational cost of HP-GP-TS?

---

> ### Author Response · Authors · 2025-11-27
> **Response to reviewer zenE**
>
> We thank **reviewer zenE** for their thoughtful review and feedback. We are delighted to see that the reviewer recognizes the experimental strengths of the proposed algorithms, in particular HP-GP-TS.
>
> > How is $\sum_{t}\sigma_{t,p^{}}^{2}(x^{*})$ bounded in Thm 4.3?
>
> We have not provided a bound for this term and it could be linear in $T$ in the worst case. We have updated section 4.1 in the paper to make this clear.
>
>
> > The UCB-based bound (Thm 4.5) includes an unbounded term representing the cost of learning the prior in the general case.
>
> We have updated section 4.2 in the paper to make it clear that we have not bounded this term. Additionally, we have added new results demonstrating the empirical behavior of the unbounded term for HP-GP-TS to appendix D. We observe that this unbounded term is empirically smaller and grows slower than the cumulative regret.
>
> > Given the $\mathcal{O}(T^4 |P|)$ complexity, are there approximations to mitigate the computational cost of HP-GP-TS?
>
> For exact GPs with a horizon of $T$, the complexity $\mathcal{O}(T^4)$ is unavoidable to our knowledge. In practice for long horizons, one can utilize approximate GP methods to mitigate this cost.
>
> As we highlight in the previous work section, plenty of previous work on Bayesian optimization have (empirically) incorporated hyperpriors with continuous distributions. Given an acquisition function for prior $p$, $\alpha_p(x)$, they compute $x_t = \arg\max_x \mathbb{E}_{p \sim P_t} [\alpha_p(x)]$ by MCMC-sampling. If MCMC is run for $L$ steps at every timestep, the complexity would be $\mathcal{O}(T^4L)$. For a continuous set of priors (such as kernel lengthscale values), HP-GP-TS could equivalently be implemented with MCMC if it's computationally cheaper than performing a fine discretization.

---

### Official Review · Reviewer_J9PU · 2025-11-07

**Soundness:** 3
**Presentation:** 3
**Contribution:** 2
**Rating:** 4
**Confidence:** 4

**Summary:**

This paper studies Bayesian optimization with an unknown prior known to be one of a finite number of candidates.  First a prior elimination + Thompson Sampling approach is given complementing an existing prior elimination + UCB approach.  Then a fully Bayesian approach is given based on Thompson sampling.  The main contributions are regret analyses for these, and experiments are also given.

**Strengths:**

The paper studies an important topic that probably deserves more attention.  The paper is well written and the mathematical analysis seems to be sound (though I didn't check the appendices closely).  One of the results hints at a result with average information gain instead of worst-case, which would be a good contribution (but see below for non-minor caveats).  The experiments are also appreciated.

**Weaknesses:**

I have potentially significant concerns with each of the the main results:
- Theorem 4.3 has an extra sum of variance terms at $x^*$ which does not appear in regular analyses.  Moreover, the authors do not comment on this extra term, and do not provide any comparison to PE-GP-UCB (where it seems there is no such term).  I suspect this term could be linear in $T$, or at least hard to prove it's sublinear.
- Similarly, Theorem 4.5 contains a sum of differences of U values that looks potentially linear in $T$ or at least hard to bound.  Again, no comment is made on this term.
- Lemma 4.4 is a bit complicated / hard to grasp, and I'm not sure of its purpose.  If the purpose is just for the 4 lines of discussion after, then I think a fair bit of editing should be done to make its meaning and role clearer.
- Theorem 4.6 looks quite weak in the sense that it matches what I would expect if the $|\mathcal{X}|$ arms were all independent without correlations.  Exploiting correlations is crucial in GP settings, and one is almost always interested in settings with much more domain points than the time horizon.

Since this paper is primarily theoretical, I view these concerns as quite significant.

Some less important notes:
- Finite $P$ is quite restrictive, but OK since handling continuous sets might be out of reach for now.
- The authors mention linear bandits, and the reader might question whether any of these can be "kernelized" (I guess not easily)
- First line of Section 4.1: I don't know what "larger" means (larger than what?)

**Questions:**

The weaknesses above have implicit questions.  The main one is -- can you prove the extra terms in Theorems 4.3 and 4.5 are sublinear, at least in certain specific scenarios?

---

> ### Author Response · Authors · 2025-11-27
> **Response to reviewer J9PU**
>
> We thank **reviewer J9PU** for their detailed review and feedback. We agree with the reviewer that the topic of the paper deserves more attention. We address the reviewer's comments below.
>
> > Theorem 4.3 has an extra sum of variance terms at $x^*$ which does not appear in regular analyses... Similarly, Theorem 4.5 contains a sum of differences of U values that looks potentially linear in T or at least hard to bound.
>
> To clarify, we have not provided a bound for these terms and they could be linear in $T$ in the worst case. We have added clarifications to section 4.1 and 4.2. Additionally, we have added new results demonstrating the empirical behavior of the unbounded term for HP-GP-TS to appendix D. We observe that this unbounded term is empirically smaller and grows slower than the cumulative regret.
>
> > Theorem 4.6 looks quite weak in the sense that it matches what I would expect if the arms $|X|$ were all independent without correlations.
>
> The information-theoretic framework ignores the structural assumptions and the regret bound is only sublinear when the number of arms is smaller than the number of time steps. We have added clarifications to this in section 4.2.
>
> > Lemma 4.4 is a bit complicated / hard to grasp, and I'm not sure of its purpose.
>
> The purpose of Lemma 4.4 is to understand the behavior of the posterior probability of the true prior, $P_{t+1}(p^{\*})$, in a simplified setting. The lemma shows that $P_{t+1}(p^*)$ depends on the whitened difference vector ${\Sigma}^{-\frac{1}{2}} \mu$. We have updated the paragraph below Lemma 4.4 to make the lemma and its interpretation more clear.
>
> > The authors mention linear bandits, and the reader might question whether any of these can be "kernelized" (I guess not easily).
>
> The algorithm of Hong et al (2022b) most closely resembles HP-GP-TS in the linear setting. As discussed in the joint response, the analysis performed by Hong et al (2022b) for the linear setting contains some non-obvious steps that are hard to motivate even in the linear setting. As such, kernelizing these results seems difficult.
>
> Joey Hong, et al. (2022b). "Thompson sampling with a mixture prior." International Conference on Artificial Intelligence and Statistics. PMLR.
>
> > Finite $P$ is quite restrictive, but OK since handling continuous sets might be out of reach for now.
>
> Indeed, continuous sets seem challenging from a theoretical perspective at the moment. From an implementation perspective, sampling from a continuous hyperposterior can be done using MCMC and has been considered by previous work (although they draw multiple samples per time step to approximate the average acquisition function). We do not expect the experimental results to meaningfully change if we had used a continuous set of priors, see our response to reviewer RXxU.
>
> > First line of Section 4.1: I don't know what "larger" means (larger than what?)
>
> By 4 larger steps, we mean that all the steps in the proof can be organized into 4 parts.

---

> > ### Comment · Reviewer_J9PU · 2025-11-28
> >
> > Thanks for the response and for adding the acknowledgments of the limitations.  My impression is still that the limitations are quite non-minor at least from a theoretical viewpoint, but I will discuss this with the other reviewers.

---

### Author Response · Authors · 2025-11-13
**Publication of code**

As discussed in the reproducibility statement, we have made the code for the experiments available in an anonymous repository [here](https://anonymous.4open.science/r/Prior-Selection-GP-Bandits-C95D).

---

### Author Response · Authors · 2025-11-27
**Joint response to all reviewers**

We would like to thank all the reviewers for their comments and feedback. There were some comments that were common among the reviewers that we address here. Additionally, **we want to highlight the updates made to the paper.** Additions to the paper have been marked with blue and text that will be removed have been marked in red.


**About the regret bounds in Theorems 4.3, 4.5, and 4.6:**

The most closely related work is that of Hong et al (2022b). The analysis performed by Hong et al (2022b) contains some non-obvious steps that we believe do not hold, in neither the linear nor GP setting. This highlights the challenges with analyzing these algorithms.

In fact, our work studies the application of the two key approaches for regret analysis in bandits, UCB-based and information theoretic, to our problem and shows the promises and challenges with each approach. The first approach leaves one term unbounded (although one would expect that they decrease over time as we observe empirically in appendix D in the updated version) and the second approach is sublinear only when $|\mathcal{X}| < T$. Moreover, in the updated version, we in detail demonstrate the issues with the most relevant work which aims to address a similar problem in a linear setting (Hong et al 2022b).


1. We have not provided a bound for the term $\sum_{t \in [T]} \sigma_{t, p^\*}^2(x^\*)$ in Theorem 4.3 nor for the term $\sum_{t = [T]} \mathbb{E}[ U_{t, p^\*}(x^\*) - U_{t, p_t} (x^\*)]$ in Theorem 4.5. **We have added clarifications to section 4.1 and 4.2 to make this more apparent. Additionally, we have added a figure demonstrating the empirical behavior of the unbounded term for HP-GP-TS to appendix D.** We observe that this unbounded term is empirically smaller and grows slower than the cumulative regret.

2. **We have added clarifications before Theorem 4.6** that the information-theoretic framework does not account for the structural assumptions and is only sublinear when the number of arms is smaller than the number of time steps, based on the feedback from reviewer J9PU and RXxU.

3. We have updated the discussion of Lemma 4.4 in section 4.2 based on the beedfack from reviewer J9PU.

4. Reviewer RXxU brought the work of Hong et al (2022b) to our attention. Hong et al propose a Thompson sampling algorithm with a mixture prior (MixTS). HP-GP-TS can be seen as an instantiation of MixTS for GP-bandits. **We have updated the discussion of previous work in section 2 to include the work of Hong et al (2022b) and its relation to our work.**

5. Hong et al (2022b) provide a general outline of a regret bound for standard bandits with bounded rewards and a detailed proof for the linear setting. However, **the proof for the linear setting contains non-obvious steps that we find difficult to motivate. We have added a discussion of these steps to appendix E and F.** To summarize, in the proof, the authors condition on the regret being bounded but for subsequent steps in the proof 1) assume that this does not affect the distribution of the optimal arm, 2) apply the boundedness assumption to the posterior mean, 3) drop the boundedness assumption without motivation, and 4) add zero-mean Gaussian inside a minimum incorrectly.
**Due to this, we do not believe that Theorem 1 in Hong et al (2022b) has been properly proven.**

6. **We have added a detailed discussion of how to apply the analysis of Hong et al (2022b) to HP-GP-TS and the challenges that arise in section F in the appendix.**

---

### Meta-Review · Area_Chair_hXBd · 2026-01-07

**Summary:**

This looks like a promising paper that needs more work before being ready for publication. In particular, all reviewers point out concerns with Thm 4.3 and 4.5 that are not easy to alleviate. Experimental evaluation of the terms in Thm.4 .3 and 4.5 are a good start but I believe that more is needed since these are central results of the paper. In particular, I would expect some theoretical insights into these terms.

**Reviewer Concerns:**

Some misunderstandings were clarified but the main concerns raised by the reviewers, the terms in Theorem 4.3 and Theorem 4.5, need a proper revision and more work.

**Reviewer Scores:**

I don't think that the reviewer scores would change: the empirical evaluations are useful but are not a replacement for theory and adding these in a rebuttal is not ideal; these should have already been included in the submission.

---

### Decision · Program_Chairs · 2026-01-26

Reject